# Black Cumin (*Nigella sativa* L.): A Comprehensive Review on Phytochemistry, Health Benefits, Molecular Pharmacology, and Safety

**DOI:** 10.3390/nu13061784

**Published:** 2021-05-24

**Authors:** Md. Abdul Hannan, Md. Ataur Rahman, Abdullah Al Mamun Sohag, Md. Jamal Uddin, Raju Dash, Mahmudul Hasan Sikder, Md. Saidur Rahman, Binod Timalsina, Yeasmin Akter Munni, Partha Protim Sarker, Mahboob Alam, Md. Mohibbullah, Md. Nazmul Haque, Israt Jahan, Md. Tahmeed Hossain, Tania Afrin, Md. Mahbubur Rahman, Md. Tahjib-Ul-Arif, Sarmistha Mitra, Diyah Fatimah Oktaviani, Md Kawsar Khan, Ho Jin Choi, Il Soo Moon, Bonglee Kim

**Affiliations:** 1Department of Anatomy, Dongguk University College of Medicine, Gyeongju 38066, Korea; hannanbmb@bau.edu.bd (M.A.H.); rajudash.bgctub@gmail.com (R.D.); binodtimalsina19@gmail.com (B.T.); yeasminakteracce@gmail.com (Y.A.M.); mahboobchem@gmail.com (M.A.); sarmisthacu@gmail.com (S.M.); diyahfatimah.oktav@gmail.com (D.F.O.); chjack@naver.com (H.J.C.); 2Department of Biochemistry and Molecular Biology, Bangladesh Agricultural University, Mymensingh 2202, Bangladesh; sohag2010bmb.sust@gmail.com (A.A.M.S.); tahmeed.hossain@bau.edu.bd (M.T.H.); tahjib@bau.edu.bd (M.T.-U.-A.); 3Department of Pathology, College of Korean Medicine, Kyung Hee University, Seoul 02447, Korea; ataur1981rahman@hotmail.com; 4Korean Medicine-Based Drug Repositioning Cancer Research Center, College of Korean Medicine, Kyung Hee University, Seoul 02447, Korea; 5ABEx Bio-Research Center, East Azampur, Dhaka 1230, Bangladesh; hasan800920@gmail.com (M.J.U.); sarkerpartha124124@gmail.com (P.P.S.); 6Graduate School of Pharmaceutical Sciences, College of Pharmacy, Ewha Womans University, Seoul 03760, Korea; 7Department of Pharmacology, Bangladesh Agricultural University, Mymensingh 2202, Bangladesh; drmsikder@bau.edu.bd; 8Department of Animal Science & Technology and BET Research Institute, Chung-Ang University, Gyeonggi-do, Anseong 17546, Korea; shohagvet@gmail.com; 9Department of Biotechnology, Bangladesh Agricultural University, Mymensingh 2202, Bangladesh; 10Division of Chemistry and Biotechnology, Dongguk University, Gyeongju 780-714, Korea; 11Department of Fishing and Post Harvest Technology, Sher-e-Bangla Agricultural University, Sher-e-Bangla Nagar, Dhaka 1207, Bangladesh; mmohib.fpht@sau.edu.bd; 12Department of Fisheries Biology and Genetics, Patuakhali Science and Technology University, Patuakhali 8602, Bangladesh; habib.332@gmail.com; 13Department of Pharmacy, Faculty of Life and Earth Sciences, Jagannath University, Dhaka 1100, Bangladesh; jahaanisrat6@gmail.com; 14Interdisciplinary Institute for Food Security, Bangladesh Agricultural University, Mymensingh 2202, Bangladesh; tania_sinthi0622@bau.edu.bd; 15Research and Development Center, KNOTUS Co., Ltd., Yeounsu-gu, Incheon 22014, Korea; mahbubs84@gmail.com; 16Department of Biochemistry and Molecular Biology, Shahjalal University of Science and Technology, Sylhet 3114, Bangladesh; bmbkawsar@gmail.com; 17Department of Biological Sciences, Macquarie University, Sydney, NSW 2109, Australia

**Keywords:** black seed, thymoquinone, nutraceutical, essential oil, molecular mechanism

## Abstract

Mounting evidence support the potential benefits of functional foods or nutraceuticals for human health and diseases. Black cumin (*Nigella sativa* L.), a highly valued nutraceutical herb with a wide array of health benefits, has attracted growing interest from health-conscious individuals, the scientific community, and pharmaceutical industries. The pleiotropic pharmacological effects of black cumin, and its main bioactive component thymoquinone (TQ), have been manifested by their ability to attenuate oxidative stress and inflammation, and to promote immunity, cell survival, and energy metabolism, which underlie diverse health benefits, including protection against metabolic, cardiovascular, digestive, hepatic, renal, respiratory, reproductive, and neurological disorders, cancer, and so on. Furthermore, black cumin acts as an antidote, mitigating various toxicities and drug-induced side effects. Despite significant advances in pharmacological benefits, this miracle herb and its active components are still far from their clinical application. This review begins with highlighting the research trends in black cumin and revisiting phytochemical profiles. Subsequently, pharmacological attributes and health benefits of black cumin and TQ are critically reviewed. We overview molecular pharmacology to gain insight into the underlying mechanism of health benefits. Issues related to pharmacokinetic herb–drug interactions, drug delivery, and safety are also addressed. Identifying knowledge gaps, our current effort will direct future research to advance potential applications of black cumin and TQ in health and diseases.

## 1. Introduction

The plant kingdom, in addition to maintaining the balance of the environment and providing life-sustaining oxygen, plays an essential role in human diets, functioning as an inevitable source of modern medicines. Plant-based foods meet basic nutritional demands, keep the body healthy, and protect against a wide range of ailments by boosting the immune system. In recent decades, the concepts of ‘nutraceuticals’ or ‘functional foods’ have become popular among health-conscious individuals, as there is a close link between a healthy diet and average life expectancy [1]. These concepts have also attracted the attention of dietitians, nutritionists, food scientists, physicians, as well as food and pharmaceutical industries. As the global market for functional foods expands, extensive research is underway to explore conventional foods with promising health benefits. Among the variety of functional food materials, minor, but indispensable ingredients, such as herbs and spices, which are mostly used as flavoring additives and preservatives, contain an abundance of biofunctional molecules [2]. Most of these culinary herbs and spices, although primarily used in cooking, are also known for their nutraceutical values, as they have enormous health-promoting potentials [3]. 

One spicy, medicinal herb is *Nigella sativa* L. (Ranunculaceae), also called black cumin or black seeds, which is famous for its culinary uses, and is historically precious in traditional medicine. Black cumin is native to a vast region of the eastern Mediterranean, northern Africa, the Indian subcontinent, and Southwest Asia, and is cultivated in many countries, including Egypt, Iran, Greece, Syria, Albania, Turkey, Saudi Arabia, India, and Pakistan. Being a panacea, black cumin, in the form of essential oil, paste, powder, and extract, has been indicated in traditional medicine for many diseases/conditions, such as asthma, bronchitis, rheumatism, headache, back pain, anorexia, amenorrhea, paralysis, inflammation, mental debility, eczema, and hypertension, to name a few [4]. These traditional uses of *N. sativa* seeds are largely attributed to their wide array of medicinal properties, including antioxidant, anti-inflammatory, immunomodulatory, anticancer, neuroprotective, antimicrobial, antihypertensive, cardioprotective, antidiabetic, gastroprotective, and nephroprotective and hepatoprotective properties [5]. Black cumin seed, particularly its essential oil, contains thymoquinone (TQ), thymohydroquinone, thymol, carvacrol, nigellidine, nigellicine, and α-hederin, which are mostly responsible for its pharmacological effects and therapeutic benefits [6]. The food value of black cumin, although less focused on in scientific literature, is by no means low, because it contains an adequate quantity of protein and fat, and an appreciable amount of essential fatty acids, amino acids, vitamins, and minerals [7]. Both active phytochemicals and the vital nutrients of black cumin contribute equally to the immunity and well-being of the human body, making this culinary herb a valuable source of nutraceuticals.

Here, we critically review the existing literature on the pharmacological properties and health benefits of black cumin and TQ and discuss the reported underlying molecular mechanisms. As the clinical application of TQ is limited by its poor bioavailability, we update the recent development of nanotechnology-based TQ delivery to overcome this limitation. We also highlight pharmacokinetic herb–drug interactions and address safety issues related to medicinal uses of black cumin.

## 2. Methodology

A literature-based search, covering research reports that have been published the last five years, was performed to retrieve information on the chemistry, health effects, molecular pharmacology, herb–drug interaction, nanotechnology-based drug delivery, and safety of black cumin and TQ from accessible online databases, such as PubMed, Web of Science, Scopus, and Google Scholar, using the key search terms of ‘black cumin’ and chemical constituents, antioxidant, anti-inflammatory, immunomodulatory, neuroprotective, nephroprotective, anti-obesity, cardioprotective, hepatoprotective, anticancer, nanotechnology or toxicity, etc. This review covers those articles that demonstrate the health benefits of black cumin alone or its compounds or both.

## 3. Evolution of Trends in Research with Black Cumin

*N. sativa* first appeared in scientific literature more than a century ago [8]. Since the publication of the first compound (melanthigenin) [9] and the first pharmacological report (antibacterial) [10,11] a couple of decades ago, the health benefits and chemical profiles of *N. sativa* were reported almost consistently. However, significant progress in research was made only in the last two decades (Figure 1A). There is an uphill trend of scientific publications (Figure 1A), indicating growing interest of the scientific community in the health-promoting potentials of *N. sativa*. Although research on *N. sativa* is ongoing in nearly half of the world, most of the research has been implemented in the Middle East and South Asia, half of which has been in Iran, Egypt, India, and Saudi Arabia (Figure 1B). Although the research articles account for the lion’s share of the total documents, the review articles share about one-tenth (Figure 1C), which is further indication of the popularity of *N. sativa* in the scientific community. Studies on *N. sativa* in various fields of medicinal research, including medicine, pharmacology, biochemistry, and chemistry provide ample evidence on the prospects of this widely popular natural product in clinical medicine (Figure 1D). Improvement in chemical profiles, agronomical traits, and adaptability of *N. sativa* through various biotechnology tools in order for enhancing its pharmacological attributes has been reported in the literature. Being highly focused, the pharmacological reports on *N. sativa* in cancer dominate those in other diseases/pathological conditions (Figure 1E). However, the publications on the protective effects of *N. sativa* in brain disorders, diabetes, cardiac problems, kidney diseases, skin diseases, and hepatic diseases are also equally significant. 

## 4. Ethnopharmacological Aspects

*Nigella sativa* L. belongs to the family Ranunculaceae; it is an annual herbaceous flowering plant, growing to a height of 20–30 cm, with linear leaves, bluish-colored flowers, capsule-shaped fruits, and small black colored seeds resembling cumin (thus referred to as black cumin). Black cumin, recognized as a Haba al-Barakah (blessed seeds) or miracle herb, was considered by ancient herbalists to be “The herb from heaven” [12]. The Prophet Mohammad (PBUH) once stated, “This black cumin is healing for all diseases except death” [13], and therefore, it is known as ‘Prophetic medicine’ in the Muslim community. The Holy Bible also mentioned black cumin for its curative properties and designated it as ‘Melanthion’ by Hippocrates and Dioscorides [14]. In his famous book ‘The Canon of Medicine’, Avicenna (Ibn Sina), a renowned physician of the 10th century and the father of early modern medicine, has highlighted several health beneficial properties of black cumin, such as enhancement of the body’s energy and recovery from tiredness and disconsolateness [5].

Black cumin has long been prescribed in the traditional systems of medicine, such as Unani, Ayurveda, Tibb, and Siddha, and is used across Arab nations, Asia, Africa, and Europe to treat various diseases and ailments, such as asthma, bronchitis, rheumatism, headache, back pain, paralysis, inflammation, and hypertension [15]. Moreover, the external application of black cumin oil has an antiseptic and local anesthetic value. The oil has also been used topically to the nasal abscesses, orchitis, swollen joints, and to treat skin conditions, such as blisters and eczema [4]. Black cumin has been found in the tomb of the Egyptian Pharaoh Tutankhamun. The ancient Egyptian civilization used black cumin as a preservative in the process of mummification [16] probably due to its antibacterial and insect repellent actions [17,18].

## 5. Phytochemical Profiles

The phytochemical composition of black cumin varies, depending on the growing regions, maturity stage, processing methods, and isolation techniques. Bioactive phytochemicals of black cumin, comprising major and minor secondary metabolites, have been categorized into different chemical classes.

### 5.1. Terpenes and Terpenoids

Thymoquinone (TQ) and its derivatives, such as carvacrol, 4-terpineol, α-pinene, thymol, t-anethol, thymohydroquinone (THQ), dithymoquinone, p-cymene, sesquiterpene longifolene, and several other compounds, constitute the terpenes and terpenoids family, which is the major chemical group of black cumin (Figure 2A). The versatility in the pharmacological characteristics of black cumin is mainly due to the presence of quinine components, the most prevalent of which is TQ. 

### 5.2. Phytosterols

Oil extracted from black cumin contains several sterols, of which β-sitosterol (44–54%) is the main sterol (Figure 2B). Comprising 16.57–20.92% of total sterols, stigmasterol constitutes the second major sterol in black cumin oil [19]. The oil also contains a smaller percentage of Δ^7^-stigmasterol, Δ^7^-avenasterol, campesterol, and cholesterol. The presence of significant levels of sterols makes black cumin an effective natural agent in lowering blood cholesterol and preventing cardiovascular diseases.

### 5.3. Alkaloids

Alkaloids of black cumin can be classified depending on the alkaloid skeletons: isoquinoline alkaloids, such as nigellicimine and nigellicimine-N-oxide, and pyrazole or indazole alkaloids, such as nigellidine and nigellicine (Figure 2C). In addition, alkaloid nigelamines A1–A5 are also reported from black cumin belonging to the diterpene family and proclaimed potent lipid metabolism-promoting activity [20] 

### 5.4. Tocols

Tocopherols are important natural antioxidants that scavenge free radicals and inhibit lipid peroxidation in biological membranes. There are four isomers of tocopherols such as alpha (α), beta (β), gamma (γ), and delta (δ), which are distinguished by the locations of methyl group on the chromanol ring (Figure 2D). Among various tocopherols, γ-tocopherol content is the highest with an amount ranging from 8.57 to 34.23 ppm [21]. The content of tocopherol isomers could be affected by methods of extractions. Differences in the content of tocopherol isomers in black cumin may arise due to the variation of the cultivated areas, maturity period, and storage conditions [22].

### 5.5. Polyphenols

As presented in Figure 2E, a total of 19 polyphenols were identified from seeds using HPLC–UV–MS [23]. These are caftaric acid, gentisic acid, caffeic acid, chlorogenic acid, p-coumaric acid, ferulic acid, sinapic acid, cichoric acid, hyperoside, isoquercitrin, rutin, myricetin, fisetin, quercitrin, quercetin, patuletin, luteolin, kaempferol, and apigenin. Of these, quercetin and kaempferol were reported to be the highest in black cumin with 105.55 ± 0.12 and 12.15 ± 0.04 μg/g dry weight of plant material, respectively. Being an antioxidant polyphenol, kaempferol helps prevent oxidative damage of cells and quercetin protects against various diseases such as osteoporosis, lung cancer, and cardiovascular problems. Moreover, kaempferol was shown to prevent arteriosclerosis by inhibiting low-density lipoprotein oxidation and platelet formation in the blood [24,25].

### 5.6. Miscellaneous Components

Black cumin has become an enriched natural product by the presence of several other chemical constituents, such as special carbohydrates (rhamnose, xylose, and arabinose), glycerolipids (monoacylglycerols, diacylglycerols, and triacylglycerols), phospholipids (phosphatidylinositol, phosphatidylcholine, and phosphatidylglycerol), vitamins (vitamin A, E and C, folic acid, thiamin, riboflavin, pyridoxine, and niacin), minerals, and some alkane hydrocarbons (n-nonane, 2-undecanone, n-octyl isobutyrate, and 8-heptadecene).

## 6. Benefits of Black Cumin on Human Health and Disease Conditions

Health benefits of black cumin and its bioactive TQ cover almost every physiological system, ranging from the nervous system to the integumentary system, and metabolic disorders, and various cancers (Tables 1–10).

### 6.1. Antioxidant Effects

Health benefits of black cumin are largely vested on its antioxidant property. Here, a summary of recent studies focused on their antioxidant properties in cell-based in vitro models and in vivo models, covering the last five years, is presented. Being a potential source of natural antioxidants, black cumin lowered the reactive oxygen species (ROS) level while upregulating antioxidant enzymes, such as superoxide dismutase (SOD) and catalase (CAT), and molecules, such as glutathione (GSH), as evident in several studies [26,27]. El-Gindy et al. reported a significant rise in blood TAC and a reduction in malondialdehyde (MDA) in rabbits supplemented with 600 mg/kg of black cumin seeds [28]. In Wister rats given with NSO (1 mL/kg), there was a significant reduction in ROS and nitrous oxide production in amygdala, thereby attenuating chlorpyrifos-induced oxidative stress [29]. TQ treatment resulted in the reduction of intracellular ROS and protection against hydrogen peroxide-induced neurotoxicity in human SH-SY5Y cells by a mechanism that involves upregulation of antioxidant related genes (SOD and CAT), as well as signaling genes, such as c-Jun N-terminal kinase (JNK), extracellular signal-regulated protein kinase (ERK)1/2, p53, protein kinase B (Akt) 1, and nuclear factor kappa-light-chain-enhancer of activated B cells (NF-κB) [30]. In adult male rats exposed to contaminated drinking water with lead acetate (2000 ppm) for five weeks, TQ (5 mg/kg/day) ameliorated toxic effects by inducing activities of CAT, glutathione reductase (GR), glutathione peroxidase (GPx), and SOD, and by increasing GSH level in liver tissues [31]. TQ treatment has also been shown to reduce oxidative stress markers (superoxide, hydrogen peroxide, and nitric oxide) and attenuate oxidative stress in lipopolysaccharide (LPS)/interferon-gamma (IFNγ) or H_2_O_2_-activated BV-2 microglia by promoting antioxidant enzymes (SOD and CAT), and GSH level, downregulating pro-oxidant genes and upregulating antioxidant genes [32]. A meta-analysis of five studies using 293 human subjects suggests that black cumin supplementation may have a beneficial role as an antioxidant by improving SOD levels without affecting MDA level and total antioxidant capacity [33]. With these recent data, it can be concluded that black cumin (in the form of NSO) and its main ingredient TQ have potential antioxidant values that underlie their protective actions against oxidative stress-induced cellular pathology. Further clinical trials are needed to determine the protective functions of black cumin and its compounds against oxidative stress-induced cellular pathology occurred in different diseases condition.

### 6.2. Anti-Inflammatory Effects

Anti-inflammatory activities are important pharmacological properties of black cumin and TQ [34]. Here, in addition to the antioxidant properties, recent developments on the anti-inflammatory potentials of black cumin seeds, covering the last five years, are focused on. In low-grade inflammation in human pre-adipocytes, freshly extracted NSO reduced the interleukin-6 (IL-6) level, while stored NSO reduced IL-1β level [35]. Following NSO treatment (400 mg/kg) in rats with carrageenan-induced paw edema, there was a significant improvement in the pro-inflammatory cytokines IL-6, IL-12, and tumor necrosis factor (TNF)-α in paw exudates and sera [36]. Moreover, topical application of balm stick containing 10% NSO in rats with paw edema substantially mitigated acute and sub-acute inflammation with a marked edema inhibition (60.64%), and a reduced leucocytes count (43.55% lower than control), and TNF-α level (50% lower than control) on the inflammation area [34]. 

As a major bioactive, TQ is the key compound responsible for the anti-inflammatory property of black cumin. Hossen et al. reported that TQ inhibited pro-inflammatory factors, including nitric oxide (NO), nitric oxide synthase (iNOS), TNF-α, IL-6, IL-1β, and cyclooxygenase (COX) 2 in LPS-stimulated murine macrophage-like RAW264.7 cells, involving a mechanism that includes the inhibition of IRAK-linked AP-1/ NF-κB pathways [37]. TQ also promoted the autophosphorylation of TANK-binding kinase 1 (TBK1), reduced the mRNA expression of interferons (IFN-α and IFN-β), and downregulated the IRF-3 signaling pathways in LPS-stimulated murine macrophage-like RAW264.7 cells [38]. Current evidence of anti-inflammatory potentials of black cumin and TQ are promising, however, most of the studies so far have been conducted in animal models. Future studies should focus on determining the anti-inflammatory potential in ameliorating human disease conditions.

### 6.3. Immunomodulatory Effects

Black cumin and TQ were shown to exert immunostimulatory functions as reported in several preclinical and clinical studies. Sheik et al. investigated the immunomodulatory effect of ethanolic extract of black cumin on murine macrophage cell line (J774A.1) and found that the extract increased macrophage population [39]. Moreover, black cumin extract has been shown to stimulate phagocytic activities of three types of macrophages [40]. Evaluating the effect of black cumin on asthma-related inflammatory mediators, Koshak and the team reported that the oily TQ-rich extract promoted immune response by reducing IL-2, IL-6, and PGE2 in primary T-lymphocytes and IL-6 and PGE2 in primary monocytes [41].

A recent study reported immunomodulatory effect of black cumin oil (NSO) in *S. typhimurium* infected rats in which the total number of leukocytes, neutrophils, eosinophils, basophils, lymphocytes, monocytes were maintained, and macrophages were activated [42]. Furthermore, the immune response was inclined against H9N2 avian influenza virus (AIV-H9N2) by the oral administration of 1% IMU (Immulant, a commercial product based on Echinacea and black cumin) after AI-H9N2 vaccination, dampening the severity of infection in stressed chicken [43]. Additionally, feeding of black cumin (2%) supplemented ration to broiler chickens exhibited a positive effect on the immune response by enhancing the antibody response against the Newcastle disease vaccine [44].

In a clinical trial on children with beta-thalassemia major, black cumin powder (2 g/day, per os (p.o.), along with foods or drinks, for three months) stimulated immune system by enhancing cluster of differentiation-4 (CD4^+^), CD8^+^ and cell counts of leucocytes (WBCs) [45]. Furthermore, NSO extracts also showed immunomodulatory effects in patients with rheumatoid arthritis by modifying subsets of T-lymphocytes [46].

### 6.4. Protection against Neurological Disorders

Black cumin and TQ have shown their therapeutic promises against a range of neurological conditions, including neurodegenerative disorders (Alzheimer’s disease (AD), and Parkinson’s disease (PD)), ischemic stroke and acute brain injury, anxiety and depression, epilepsy, and schizophrenia (Table 1). Moreover, black cumin and TQ were shown to protect against various chemical-induced neuronal injury in experimental conditions (Table 1). The neuroprotective potentials of black cumin and TQ mostly stem from antioxidative and anti-inflammatory properties [47] (Figure 3).

#### 6.4.1. Protection against Neuroinflammation

Neuroinflammation is one of the main contributing factors to the pathogenesis of major neurodegenerative diseases [48]. Microglial activation in response to stimuli is associated with the progression and ignition of neuroinflammation. The activation of NF- κB family of transcription factors upon binding with DNA is a key step in the regulation of pro-inflammatory cytokine expression as activated NF-κB induces transcription of genes encoding pro-inflammatory cytokines, chemokines, and pro-inflammatory enzymes. Thus, the pharmacological intervention targeting microglial activation has significant therapeutical value in inflammation-mediated neuronal pathobiology. In LPS/IFNγ or H_2_O_2_-activated BV-2 microglial cells, ΤQ treatment (12.5 μM for 24 h) attenuated oxidative stress and inflammation by increasing GSH, SOD, and CAT, and by reducing lipid hydroperoxides (LPO), cytokines (IL-2, IL-4, IL-6, IL-10, and IL-17a), and chemokines (CXCL3 and CCL5) [32,49]. Besides, TQ prevented LPS-activated neuroinflammation in BV2 microglia by activating liver kinase B1 (LKB1), AMP-activated protein kinase (AMPK) and Sirtuin 1 (SIRT1) [50]. The overall anti-inflammatory activity of black cumin, especially the attenuating effect of TQ on neuroinflammation, makes this natural product a suitable therapeutic agent to be used against inflammation-mediated neurological disorders.

**Table 1 nutrients-13-01784-t001:** Comprehensive summary on the protective effects of black cumin against neurological and mental problems.

Treatment with Doses	Experimental Model	Major Findings (Including Molecular Changes)	References
Neuroinflammation			
TQ (12.5 μM for 24 h)	LPS/IFNγ or H_2_O_2_-activated BV-2 microglial cell	↓H_2_O_2_; ↑GSH; ↑SOD and CAT	[32]
TQ (12.5 μM for 24 h)	LPS/IFNγ or H_2_O_2_-activated BV-2 microglial cell	↑Glutaredoxin-3, biliverdin reductase A, 3-mercaptopyruvate sulfurtransferase, and mitochondrial Lon protease; ↓IL-2, IL-4, IL-6, IL-10, and IL-17a, CFB, CXCL3 and CCL5	[49]
TQ (2.5–10 μM)	LPS-activated neuroinflammation in BV-2 microglial cell	↓ROS; ↑LKB1 and AMPK; ↑nuclear accumulation of SIRT1	[50]
Alzheimer’s disease			
TQ (100 nM)	Aβ1–42-induced neurotoxicity in hiPSC-derived cholinergic neurons	↑GSH; ↓ROS; ↓synaptic toxicity, attenuate cell death and apoptosis	[51]
TQ fraction rich nanoemulsion of seeds (TQRFNE) (250 and 500 mg/kg BW)	High fat/cholesterol diet-induced neurotoxicity in rats	↓Aβ40 and Aβ42; ↑APP; ↓PSEN1 and PSEN2; ↓BACE1 and RAGE; ↑IDE and LRP1	[52]
TQ fraction rich nanoemulsion of Nigella seeds (TQRFNE) (250 and 500 mg/kg BW)	High fat/cholesterol diet-induced neurotoxicity in rats	↓Memory impairment; ↓lipid peroxidation and soluble Aβ levels; ↑total antioxidant status and antioxidants genes expression	[53]
TQ (10, 20, and 40 mg/kg/day p.o. for 14 days)	Combined AlCl_3_andD-Gal-induced AD in rats	Improved cognitive deficits; ↓Aβ formation and accumulation; ↓TNF-α and IL-1β; ↓TLRs pathway components; ↓NF-κB and IRF-3 mRNAs	[54]
TQ (intragastrically, 20 mg/kg/day once daily for 14 days)	Combined AlCl_3_ and D-Gal induced neurotoxicity in rats	↑ Memory performance; ↑ SOD; ↓TAC; ↓MDA; ↓NO; ↓TNF-α; ↓AChE activity; ↑BDNF and Bcl-2	[55]
TQ (intragastrically, 20 mg/kg/day for 15 days)	Aβ (1–42) infused rat model of AD	↓Memory performance (Morris water maze test); ↓IFN-γ; ↑ DCX and MAP2	[56]
Parkinson’s disease			
TQ (100 nM)	α-Synuclein-induced rat hippocampal and hiPSC-derived neurons	↑Synaptophysin; ↓synaptic vesicle recycling; ↑spontaneous firing activity	[57]
TQ (10 mg/kg BW, 1 week prior to MPTP at 25 mg/kg BW)	MPTP-induced mouse PD model	↓MDA; ↑GSH; ↑SOD; ↑CAT; ↓IL-1β and IL-6; ↓TNF-α; ↓COX-2 and iNOS; ↓α-synuclein aggregation	[58]
TQ (7.5 and 15 mg/kg/day, p.o.)	Rotenone-induced rat PD model	↓Oxidative stress; ↑Parkin; ↓ Drp1; ↑dopamine; ↑TH levels	[59]
Ischemic stroke			
Hydroalcoholic seed extract (20 mg/kg BW)	Global ischemia in rats	↓Brain edema and infarct volume; ↑VEGF, HIF and MMP9	[60]
TQ	Stroke-prone spontaneously hypertensive rats	↓Chemoattractant protein-1, Cox-2, IL-1β, and IL-6	[61]
Traumatic brain injury			
TQ (5 mg/kg/day for seven days)	Feeney’s falling weight-induced moderate head trauma	↑Neuron densities; ↓MDA	[62]
Anxiety and Depression			
Ethanolic seed extract	Chronic stress-induced depression model	↓NO	[63]
TQ-loaded solid lipid nanoparticle (20 mg/kg, p.o.) and TQ (20 mg/kg, p.o.)	Chronic stress-induced depression model	↓IL-6, TNFα; ↑BDNF; ↑5-HT; ↑IDO	[64]
NSO (0.2 mL/kg for 20 days)	Stress-induced depression model	↑Memory performance (FST)	[65]
Hydroalcoholic seed extract (200 and 400 mg/kg)	Stress-induced depression and anxiety model	↑Anxiolytic (Open field and elevated plus-maze test); ↓depression (FST)	[66]
Epilepsy			
Ethanolic seed extract (400 mg/kg/day, p.o.)	PTZ-induced kindling mode	↓Kindling development; ↑memory performance (Morris water maze test); ↓LTP	[67]
NSO (400 and 600 mg/kg BW)	Electroshock-induced seizures	↑Anticonvulsant activity	[68]
TQ (10 mg/kg, i.p)	Lithium chloride and pilocarpine-induced seizure	↑Memory performance; ↑SOD; ↑Nrf2, HO-1	[69]
TQ (10 mg/kg, i.p)	Lithium chloride and pilocarpine-induced seizure	↑Memory performance; ↓COX-2, TNF-α and NF-κB	[70]
Hydroalcoholic seed extract (200 and 400 mg/kg for 5 days)	PTZ-induced seizure model	↑Memory performance (Morris water maze and passive avoidance test); ↑ total thiol; ↓MDA	[71]
Schizophrenia			
TQ (20 mg/kg, daily for 28 days, i.p.)	Mice model of schizophrenia (haloperidol-induced catalepsy and apomorphine-induced climbing behavior)	Anti-amnesic effect; ↓AChE activity; ↓ TBARS; ↑GSH and catalase; ↑dopamine level	[72]
Miscellaneous effects			
Chemical-induced toxicity			
TQ (5 mg/kg, i.p. for 11days)	Acrylamide-induced neurotoxicity in rats	Improved gait abnormalities; ↑GSH; ↓MDA;↓caspases 3 and 9, and Bax/Bcl-2, pP38/P38 and pJNK/JNK; ↓pERK/ERK; restore BBB integrity	[73]
TQ (5 and 10 mg/kg, i.p., for 11 days)	Acrylamide-Induced Peripheral Nervous System Toxicity in rats	Improved gait abnormalities; ↑GSH and ↓MDA;↓caspases 3 and 9, and Bax/Bcl-2, pP38/P38 and pJNK/JNK; ↓pERK/ERK	[74]
TQ (10 µM and 20 µM)	Arsenic-induced cytotoxicity in SH-SY5Y cells	Promotes DNA repairing; ↓ROS, balanced transmembrane potential; ↓ Bax and PARP-1, and ↑Bcl-2	[75]
TQ (5 mg/kg/day, for 3 days, p.o.)	Arsenic-induced hippocampal toxicity in rats	Improve anxiety behavior (Open field test and elevated plus maze); ↑GSH and SOD; ↓DNA damage; ↓TNF-α and INF-γ	[76]
TQ (2.5 and 5 mg/kg BW, for 8 days, p.o.)	Arsenic-induced hippocampal toxicity in Wistar rats	↑Δψm	[77]
NSO (1 mL/kg BW for 7 days)	Dichlorvos-induced oxidative and neuronal damage in rats	↓Vacuolation in the frontal and cerebellar cortices;↑TAC and GSH↓ROS	[78]
Radiotoxicity			
TQ	Radiation-induced oxidative stress in brain tissue	↑Antioxidant enzymes	[79]

p.o.: per os; BW: body weight.

#### 6.4.2. Protection against Alzheimer’s Disease

Alzheimer’s disease (AD), the most prevalent age-associated neurodegenerative disorder with cognitive decline, is characterized by the presence of intracellular neurofibrillary tangles, consisting of hyperphosphorylated tau protein, and extracellular senile plaques composed of amyloid-beta (Aβ) peptide [80]. These two major pathological hallmarks contribute to changes in numerous cellular and subcellular events, such as mitochondrial dysfunction, redox imbalance, and neuroinflammation that lead to activation of neurotoxic cascade towards cell death, thereby impairing synaptic communication [81,82]. These pathological events ultimately cause neurodegeneration [83].

Black cumin and TQ were shown to confer neuroprotection and improved disease outcomes in AD animal models. For example, in Aβ1–42 induced neurotoxicity, TQ protected cholinergic hiPSC neurons by restoring intracellular antioxidant levels and by inhibiting ROS generation, cell death, and apoptosis [51], and subsequently reduced quantal size of ready to release synaptic vesicle pools caused by Aβ1–42 [51]. Furthermore, in high fat/cholesterol diet (HFCD)-induced rats, TQ rich fraction nanoemulsion (TQRFNE) of black cumin decreased the brain Aβ fragment length 1–40 and 1–42 (Aβ40 and Aβ42) by modulating APP processing enzymes such as BACE1, PSEN1 and PSEN2 [52]. TQRFNE also promoted transportation activity-induced Aβ degradation via enhancement of insulin-degrading enzyme (IDE), Low-density lipoprotein receptor-related protein 1 (LRP1) and receptor for advanced glycation end-products (RAGE) enzymes [52]. In another study by the same group, supplementation of TQRFNE to Sprague–Dawley rats ameliorated HFCD-fed-induced hypercholesterolemia, memory impairment, and lipid peroxidation by improving total antioxidant status and antioxidants gene expression [53]. In aluminum chloride (AlCl_3_) and D-galactose (D-Gal) induced AD rats, treatments of TQ (10, 20, and 40 mg/kg/day for 14 days) ameliorated cognitive decline in AD rats by decreasing Aβ formation and accumulation and by inhibiting inflammatory response through the downregulation of NF-κB pathway [54]. In addition, continuous treatment of TQ (20 mg/kg/day, intraperitoneally, for 42 days) ameliorated D-Gal/AlCl_3_-induced cognitive deficit by protecting against oxidative stress, enhancing cholinergic function and increasing brain-derived neurotrophic factor (BDNF) and Bcl-2 levels [55]. Furthermore, black cumin oil was shown to protect against chronic brain hypoperfusion-induced neurodegeneration in rat’s hippocampal pyramidal CA1 neurons [84]. In Aβ (1–42) infused AD rats, TQ (20 mg/kg/day for 15 days) improved memory performance, attenuated IFN-γ expression, and increased neuronal growth-related proteins, such as doublecortin (DCX) and microtubule-associated protein (MAP2) [56]. With these and other evidence reviewed elsewhere [47,85,86,87], it can be assumed that black cumin and TQ could be potential anti-AD agents. However, clinical trials in human subjects are warranted to translate the preclinical findings of this natural product into patient use.

**Figure 3 nutrients-13-01784-f003:**
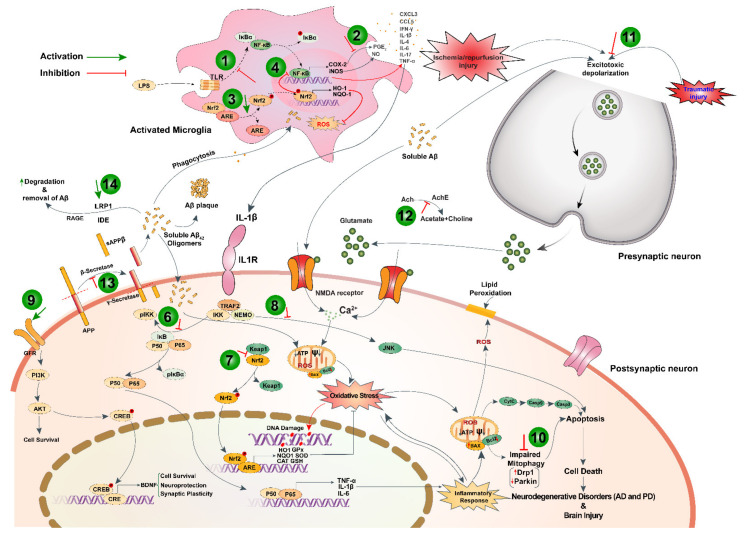
A schematic diagram illustrating the pathobiology of degenerative brain disorders and post-ischemic/traumatic consequences showing point of action of black cumin and TQ. The neuroprotective mechanisms of black cumin and TQ involve (1) attenuation of inflammatory response via inhibition of NF-κB signaling; (2) inhibition of COX-2 activity; (3) induction of antioxidant defense system via activation of Nrf2/ARE pathway; (4) cross-talk between Nrf2 and NF-κB; and (5) attenuation of oxidative stress in activated microglia; (6) protection against neuroinflammation by inhibiting NF-κB signaling; (7) priming of antioxidant defense system by activating Nrf2/ARE pathway; (8) prevention of apoptosis via downregulating pro-apoptotic JNK/Erk pathway; (9) activation of BDNF-dependent pro-survival pathway via inducing PI3K/Akt signaling; and (10) induction of mitophagy in neuron; (11) attenuation of I/R-injury via preventing excitotoxic depolarization in presynaptic terminal of neuron; (12) anticholinesterase activity; (13) anti-amyloidogenesis via blocking β-secretase activity; and (14) Aβ-clearance by upregulating IDE, LRP1, and RAGE. TLR, toll-like receptor; LPS, lipopolysaccharide; NF-κB (p50-p65), nuclear factor kappa-light-chain-enhancer of activated B cells; Nrf2, nuclear factor erythroid 2-related factor 2; ARE, antioxidant response element; IkB, inhibitor of NF-κB; IKK, IκB kinase; Keap1, Kelch-like ECH-associated protein 1; COX2, cyclooxygenase 2; iNOS, inducible isoform of Nitric oxide synthase; ROS, reactive oxygen species; HO-1, heme oxygenase-1; NQO-1, NAD(P)H quinone oxidoreductase 1; PGE2, prostaglandin E2; NO, nitric oxide; IL-1β, interleukin-1β; IL1R, interleukin-1 receptor; APP, amyloid precursor protein; LRP1; Low-density lipoprotein receptor-related protein 1; IDE, insulin-degrading enzyme; RAGE, Receptor for advanced glycation end-products; JNK, c-Jun N-terminal kinases; GluN2B, N-methyl D-aspartate receptor subtype 2B; GFR, growth factor receptor; PI3K, phosphoinositide 3-kinases; Akt, protein kinase B; CREB, cAMP-response element binding protein; BDNF, Brain-derived neurotrophic factor; Drp1; dynamin-related protein-1; AChE, acetylcholinesterase; Ach, acetylcholine; ψ, mitochondrial membrane potential. This image is modified from [88].

#### 6.4.3. Protection Against Parkinson’s Disease

Parkinson’s disease (PD), the second most prevalent neurodegenerative disorder with cognitive and motor deficits, is characterized by a gradual loss of dopaminergic neuron in the substantia nigra pars compacta. Like Alzheimer’s disease, PD-associated changes and symptoms can also be improved by TQ in experimental conditions. For instance, TQ attenuated α-synuclein-induced decrease in synaptophysin expression and increase in synaptic vesicle recycling, and thus protected against synaptic toxicity in rat primary hippocampal and hiPSC-derived neurons [57]. Ardah and the team reported that TQ protected against MPTP-induced oxidative stress and inflammatory response in PD mice [58]. With a similar neuroprotective and antioxidant mechanism, TQ attenuated rotenone-induced motor defects and alteration in the level of parkin, dynamin-related protein 1 (Drp1), dopamine, and tyrosine hydroxylase (TH) in PD rats [59]. Along with other reports reviewed elsewhere [47,86,87], this evidence suggests anti-Parkinson’s potential of black cumin, especially its constituent TQ. However, clinical trials are essential to validate the preclinical findings of black cumin and to recommend patient use.

#### 6.4.4. Protection Against Ischemic Stroke and Traumatic Brain Injury

Ischemic stroke, a pathological condition caused by a sudden cessation of blood supply to the brain, is the second leading cause of death and disability worldwide. The ischemia/reperfusion (I/R) injury that follows cerebral occlusion can initiate a cascade of events including redox imbalance, excitotoxicity, dysregulated inflammatory response, mitochondrial dysfunction, and apoptosis which ultimately lead to a neuronal loss [89].

Black cumin and TQ showed therapeutic promises against ischemic stroke and traumatic brain injury as presented in several reviews [47,87]. Soleimannejad and colleagues reported that black cumin seed extract at 10 and 20 mg/kg improved global ischemia outcomes by a mechanism that involved an increase in gene expression of vascular endothelial growth factor (VEGF), hypoxia-inducible factor-1 (HIF), and matrix metallopeptidase (MMP9) [60]. In stroke-prone spontaneously hypertensive rats, TQ improved blood pressure and ameliorated memory and cognition deficits [61].

Like ischemic stroke, traumatic brain injury is another leading cause of morbidity and mortality worldwide [90]. Following Feeney’s falling weight-induced moderate head trauma in rats, TQ treatment (5 mg/kg/day for seven days) exerted a healing effect by increasing the neuronal density in the affected areas and lowering lipid peroxidation, although the antioxidant enzyme levels were unchanged [62]. In both ischemic and traumatic brain injuries, there is a lack of human clinical trials and, therefore, it is recommended to validate the preclinically reported neuroprotective effects of black cumin and TQ.

Apart from these neuroprotective effects, black cumin seeds and TQ also protected against neurotoxicity induced by various chemicals, such as chlorpyrifos [29,91], dichlorvos [78], acrylamide [73,74], and arsenic [75,76,77], among many others. Individuals exposed naturally to these environmental toxins can develop a variety of neurological diseases and mental illnesses. Due to antidotal effects, black cumin, and TQ may be effective in preventing these neurological problems.

#### 6.4.5. Protection against Anxiety and Depression, Epilepsy, Schizophrenia, and Other Miscellaneous Neurological Problems

As presented in Table 1, black cumin and TQ show promise for their antidepressant, anxiolytic [63,64,65,66], and anti-schizophrenic [72] agents. In several animal models of epilepsy, black cumin and TQ were shown to reduce convulsion and improve memory performance. Vafaee and colleagues reported that oral administration of hydroalcoholic seed extract attenuated convulsion and improved memory performance in the pentylenetetrazole (PTZ)-induced seizure model through modulating redox status [71]. In another study, black cumin and probiotic supplementation conferred protection against seizure, seizure-induced cognitive impairment, and hippocampal long-term potentiation in PTZ-induced kindled rats [67]. NSO exhibited anticonvulsant effects in electroshock-induced seizures in rats [68]. In their two successive studies, Shao and the team reported that TQ attenuated convulsion and improved learning and memory function in lithium-pilocarpine induced models of status epilepticus via anti-inflammatory (NF-κB) [70] and antioxidant (Nrf2/HO-1) [69] signaling pathways. TQ was also shown to mitigate various types of neuropathic pain [92,93] because of the antioxidant, anti-inflammatory, anti-apoptosis, and neurotrophic properties of black cumin.

### 6.5. Anti-Cancer Effects

Black cumin and its compounds are widely known for their potent anticancer actions. Accumulating evidence suggests that chemical constituents of black cumin seeds are chemopreventive and potent in inhibiting cell proliferation and provoking apoptosis (Table 2). In a recent study, administration of black cumin seed ethanolic extract (250 mg/kg; p.o. for 5 days) was reported to attenuate diethylnitrosamine (DENA)-induced liver carcinogenesis and reduce serum AFP and VEGF levels and liver HGFβ protein in rats [94].

**Table 2 nutrients-13-01784-t002:** Comprehensive summary on the anticancer effects of black cumin.

Treatment with Doses	Experimental Model	Major Findings (Including Molecular Changes)	References
Seeds incorporated silver nanoparticles (NS-AgNP) (25–200 µg/mL)	Human breast cancer cell line (HCC-712)	Dose-dependent cytotoxicity; ↓cell density	[95]
Aqueous seed extract (11.5 µg/mL)	Human breast cancer cell line (MCF-7)	Potent cytotoxic effect with IC_50_ 11.5 µg/mL; ↑caspase-3,8 and 9, and Bax	[96]
NSO nanoemulsion (10–100 µL/mL)	Human breast cancer cell line (MCF-7)	↓Cell proliferation; ↑apoptosis and necrosis	[97]
TQ (25 µmol/L)	Human breast cancer cell line (MCF-7)	Inhibit tumor cell growth; ↑p53; induce apoptosis	[98]
Seeds incorporated platinum nanoparticles (NS-PtNP) (25, 50, 100 and 150 µg/mL)	HeLa cervical cancer and MDA-MB-231 breast cancer cell lines	Dose-dependent cytotoxic effect with IC_50_ value 36.86 µg/mL (MDA-MB-231) and 19.83 µg/mL (HeLa), respectively	[99]
TQ (0.78 µM)	HeLa cervical cancer cell line	Dose-dependent antiproliferative effect	[100]
TQ (2, 4, 6 and 8 µM)	Human colon cancer cell line (LoVo)	Inhibit metastasis; ↑JNK, p38; ↓P13K, ERK1/2, IKKα/β and NF-κB	[101]
TQ (20 µmol/L)	Human colon cancer cell line (LoVo)	Reduce cell proliferation; ↓p-P13K, p-Akt, p-GSK3β, β-catenin and COX-2; ↓PGE2, LEF-1 and TCF-4	[102]
TQ (10–120 µmol/L)	Human bladder cancer cell lines (253J and T24)	Inhibit proliferation and metastasis; ↓MYC, Axin-2, MMP-7, MET and cyclin-D1; ↓Wnt/β-catenin signaling cascade	[103]
TQ (40, 60 and 80 µM)	Human bladder cancer cell lines (253J and T24)	Significant cytotoxicity and reduction in cell proliferation; ↑caspase-3, cleaved PARP, Bax, cyt c and AIF; ↑ER-stress marker GRP78, IRE1, ATF6, ATF4 and CHOP; ↓Bcl-2 and Bcl-xl; induce apoptosis	[104]
TQ (10–50 µM)	Pancreatic ductal adenocarcinoma cell lines (AsPC1 and MiaPaCa-2)	Inhibit cell viability; reduce tumor size; ↑p53, p21; ↓Bcl-2 and HDAC; induce apoptosis and G2 cell cycle arrest	[105]
TQ (0.5–20 µM)	Human renal tubular epithelial cell line (HK2) and human renal cancer cell lines (769-P and 786-O)	Inhibit metastatic phenotype and epithelial-mesenchymal transition; ↑E-cadherin; ↓Snail, ZEB1 and vimentin; ↑LKB1/AMPK signaling	[106]
TQ (0–100 µmol/L)	Human renal cancer cell lines (ACHN and 786-O)	Inhibition of metastasis; ↑LC3; ↑AMPK/mTOR signaling; induce autophagy	[107]
TQ (40 and 50 µM)	Human kidney cancer cell lines (A498 and Caki-1)	Anti-proliferative effects with GI_50_ value 40.07 µM (A498) and 51.04 µM (Caki-1), respectively; ↑Bax; ↓Bcl-2 and p-Akt; induce apoptosis	[108]
Hexanic seed extract (0–150 µg/mL)	Human ovary cancer cell line (A2780)	Strong cytotoxic activity of SF2 with IC_50_ 10.89 µg/mL; ↑caspase-3 and 9; ↓MMP; induce apoptosis	[109]
Seed extract and NSO with OM-90(0.5 and 2.4 mg/mL)	AGS human gastric adenocarcinoma cell line	Activates mitochondrial pathways; induce apoptosis	[110]
TQ (0.1–30 µM)	Human prostate cancer cell lines (PC3 and DU145)	Inhibit metastatic phenotype and epithelial-mesenchymal transition; ↓TGF-β, Smad2 and Smad3	[111]
TQ (0–80 µM)	Head and neck squamous cells carcinoma cell lines (SCC25 and CAL27)	Dose-dependent cytotoxicity with IC50 value 12.12 µM (CAL27) and 24.62 µM (SCC25), respectively; induce apoptosis	[112]
TQ + Resveratrol (46 µM)	Hepatocellular carcinoma cell line (HepG2)	Significant cell inhibition; ↑caspase-3; ↓GSH and MDA; induce apoptosis	[113]
NSO (50–250 µg/mL)	Human liver cancer (HepG2), human breast cancer (MCF-7), human lung cancer (A-549) and normal human embryonic kidney (HEK293) cell lines	High cytotoxic effect in HepG2 cells with IC_50_ 48µg/mL; ↑ROS and LPO; ↓GSH and MMP; ↑p53, caspase-3 and 9, Bax; ↓Bcl-2; induce apoptosis	[114]
TQ (In vitro: 1–50 µMIn vivo: 20 and 100 mg/kg for 3 days; i.v.)	TNBC cells and orthotopic TNBC xenograft mice model	Inhibit cell proliferation, migration and invasion; ↓tumor growth; ↓eEF-2K, Src/FAK and Akt	[115]
TQ + Paclitaxel (In vitro: 0–100 µM In vivo: 2.4 mg/kg/day for 12 days; i.p)	Mouse breast cancer cell line (4T1) and EAC cells-induced female Balb/c mice model	Dose-dependent cytotoxicity; ↑caspase-3,7 and 12, PARP; ↓p65, p53 and Akt1; ↓JAK-STAT signaling	[116]
Ethanolic seed extract (250 mg/kg/day for 5 days, p.o.)	Diethyl nitrosamine-induced hepatocarcinogenesis in Wistar rat model	Antiangiogenic effect; ↓serum VEGF and AFP levels, and liver HGFβ level	[94]
Ethanolic seed extract and TQ (150, 250 and 300 mg/kg (extract) 6 days/week and 20 mg/kg (TQ) for 3 days/week, p.o.)	Diethyl nitrosamine-induced hepatocellular carcinoma in albino-Wistar rat model	Reduction in cell proliferation; ↑Antioxidant activity; ↓PCNA, c-fos, Bcl-2; ↓EGFR/ERK1/2 signaling	[117]
TQ + 5-fluorouracil (35 mg/kg/day for 3 days/week for 9 weeks; p.o.)	Azoxymethane-induced colon cancer in Wistar rat model	Subdues tumor growth; ↑TGF-β1, TGF-β/RII, Smad4, DKK-1, CDNK-1A and GPx; ↓Wnt, β-catenin, NF-κB, VEGF, COX2, iNOS and TBRAS	[118]
TQ + Piperine (10 mg/kg/day for 14 days; i.p)	EMT6/P cells- inoculated Balb/c mice	Inhibit angiogenesis; ↓Tumor size; ↑serum INF-ᵧ level; ↓VEGF; induce apoptosis	[119]
TQ + Resveratrol (50 mg/kg/day for 14 days; i.p)	EMT6/P cells- inoculated Balb/c mice	Inhibit angiogenesis; ↓Tumor size; ↑serum INF-ᵧ level; ↓VEGF; induce apoptosis	[120]

By virtue of its antioxidant potential, TQ, the major bioactive of black cumin, was identified to regulate diverse signaling systems in inhibiting cancer progression (Figure 4) [121]. For example, Shahin et al. reported that administration of seed extract (150, 250, and 350 mg/kg, p.o. daily for 12 days) and TQ (20 mg/kg, p.o., three alternative days/week for 12 days) exerted chemopreventive efficacy by improving antioxidant status (GSH, GST, GPx, and SOD), downregulating expressions of Bcl-2, c-fos, and PCNA and by inhibiting epidermal growth factor receptor (EGFR)/extracellular signal-regulated kinase (ERK)1/2 signaling pathway [117]. In addition, TQ (20 and 100 mg/kg) was reported to inhibit eEF-2K expression and suppress tumor growth and progression in an orthotopic TNBC xenograft mouse model [115].

TQ was shown to synergize anticancer activity of several standard chemotherapeutic drugs as well as natural chemopreventive molecules. For example, TQ (2.4 mg/kg/day; i.p. for 12 days) increased chemosensitivity of paclitaxel in Ehrlich ascites carcinoma model, where it promoted expression of tumor suppressor genes (p21, Brca1 and Hic1), induction of apoptosis-related markers (cleaved caspase-3,7 and 12 and PARP), modulation of p53 and JAK-STAT signaling and reduction of phosphorylated p65 and Akt [116]. TQ (35 mg/kg/day, p.o., 3 days/week for 9 weeks) enhanced cytotoxicity of 5-fluorouracil in AOM-instigated colon cancer rats by reducing expressions of Wnt, β-catenin, NF-κB, VEGF, COX2, iNOS, and TBRAS and upregulating expressions of TGF-β1, TGF-β/RII, Smad4, DKK-1, cyclin-dependent kinase inhibitor (CDKN)1-A, and GPx [118]. TQ (50 mg/kg/day; i.p. for 14 days) also enhanced chemosensitivity of resveratrol in the breast cancer model animals by upregulating serum INF-γ levels and the downregulation of VEGF expression [120]. Talib et al. showed that TQ (10 mg/kg/day; i.p. for 14 days) and piperine combination therapy significantly reduced tumor size and induced apoptosis by downregulating VEGF levels and elevating IFN-γ and IL-2 levels [119]. Along with these preclinical animal studies, a growing number of in vitro studies also support the anticancer potentials of black cumin and TQ, as illustrated in Table 2.

### 6.6. Anti-Obesity and Anti-Dyslipidemic Effects

Obesity or dyslipidemia has a linear relationship with cardiovascular and cerebrovascular diseases, thereby increasing the risk of mortality. Black cumin and its various preparation have been studied to explore novel therapeutic agents from natural products combating obesity and dyslipidemia.

A study with hyperlipidemic rats by Ahmed and co-investigators showed that dietary supplementation of black cumin seed extracts improved hyperlipidemic conditions by elevating high-density lipoproteins (HDL) level and lowering cholesterol, triglycerides, and low-density lipoproteins (LDL) levels [122]. Moreover, due to the presence of essential fatty acid, black cumin itself can contribute to improving dyslipidemia and the associated complications. 

In several clinical trials (Table 3), including those on hyperlipidemic patients with a smoking habit [123], menopausal women with metabolic diseases [124], outpatients of metabolic syndrome [125], and patients bearing Hashimoto’s thyroiditis [126], administration of black cumin seeds in different formulations significantly improved lipid profile and blood sugar. Along with the preclinical findings, observations from these clinical studies suggest that black cumin could be a promising candidate for anti-obesity and anti-hyperlipidemic agents. Future studies should focus on the biochemical and molecular mechanisms of black cumin seeds and their formulations for a better understanding of their anti-obesity properties.

### 6.7. Anti-Diabetic Effects

Diabetes mellitus is a chronic metabolic disorder characterized by hyperglycemia over an extended period with the abnormalities of carbohydrate, fat, and protein metabolism in the body, primarily due to the disturbance of insulin secretion and/or action. The pathobiology of diabetes is closely related to deregulated inflammation, increased oxidative stress via impaired redox homeostasis, and imbalanced blood lipid profiles [127]. The complications associated with diabetes include neuropathy, nephropathy, and retinopathy. 

Traditional medicine is a relatively affordable health care system in treating patients with diabetes. Being a part of traditional medicines, black cumin and essential oil (NSO) have great potentials as anti-diabetic agents (Table 4). Extracts of black cumin have been shown to improve disease outcomes in either alloxan or streptozotocin-induced diabetic rats or mice through the mechanism that involved attenuation of oxidative stress by increasing the activity of antioxidant enzymes [128], regulation of blood lipid profiles [129], amelioration of endothelial dysfunction [130], and enhancement of tissue regeneration and wound healing [131]. Moreover, silver nanoparticles-based green synthesis from black cumin seed extract was shown to ameliorate STZ-induced diabetic neuropathy in rats through inhibiting inflammatory signaling, restoring the antioxidant system, and increasing nerve growth factor in the brain [132].

In addition, NSO, the principal functional preparation of black cumin showed ameliorating effects in several STZ or alloxan-induced diabetic models. The major pharmacological mechanisms include the promotion of pro-survival and antiapoptotic signals [133,134], activation of growth factor signaling and inhibition of inflammatory response [135], increased insulin secretion and antioxidant molecules [136], suppression of extracellular matrix gene expression [137], and regulation lipid profiles [138]. Recently an in vitro cell-free assay reported alpha-amylase inhibitory activity of silver nanoparticles prepared from NSO, supporting its hypoglycemic effects [139].

Three clinical trials with patients suffering from type 2 diabetes (T2D) or diabetic nephropathy have been conducted, until today, to evaluate the therapeutic roles of black cumin seeds or NSO. In a randomized clinical trial on patients with diabetic nephropathy accompanying chronic kidney disease, following NSO treatment, there was a significant reduction in blood glucose, serum creatinine, blood urea, and 24 h total urinary protein levels and an increase in glomerular filtration rate, 24 h total urinary volume, and hemoglobin level [140]. In a double-blind randomized clinical trial on T2D patients, NSO supplementation was accompanied by a lowering effect on lipid profile, glycemia, C-reactive protein level, and lipid peroxidation [141]. Complying with preclinical studies, a single-blind nonrandomized controlled clinical trial on T2D patients given with black cumin seed capsule demonstrated a significant decline in TC, LDL-C, TC/HDL-C, LDL-C/HDL-C ratios, DBP, MAP and HR, and increase of serum HDL-C level [142]. As shown in Table 4, observations from various preclinical and clinical studies have suggested that black cumin and NSO could be effective herbal medicines for the management of patients with diabetic complications.

### 6.8. Cardioprotective and Antihypertensive Effects

Having cytoprotective actions against numerous adverse stimuli, black cumin seed exhibits cardioprotective effects (Table 5). For instance, post-conditioning with black cumin improved cardiac functions against I/R-induced cardiac injury by mitigating oxidative stress [143]. Pretreatment with black cumin seed ethanolic extract (800 mg/kg) exhibited a promising cardioprotective action against isoproterenol-induced myocardial infarction by improving cardiac biomarkers and antioxidant status [144].

Hypertension is a leading cause of cardiovascular disease and associated death worldwide. Black cumin seed has also shown promise in lowering hypertension (Table 5). In an angiotensin II-induced hypertensive rat model, black cumin seed has a probable normalizing effect on hypertension by antagonizing the cardiovascular effects of angiotensin II [145]. However, in a randomized controlled clinical trial in elderly patients with hypertension, black cumin administration (300 mg black cumin seed extract twice daily for 28 days) showed a slight but insignificant reduction in blood pressure [146]. On the contrary, a clinical study on patients with mild–moderate hypertension using black cumin virgin oil demonstrated positive outcomes on lipid profiles and blood pressure [147]. With the evidence stated here and in the previous sections, it can be anticipated that black cumin can be a promising candidate for developing therapeutics against heart diseases, provided that the controversies raised in some studies are properly resolved by employing appropriate experimental models.

**Table 5 nutrients-13-01784-t005:** Comprehensive summary on the cardioprotective and antihypertensive effects of black cumin.

Treatment with Doses	Experimental Model	Major Findings (Including Molecular Changes)	References
Hydroalcoholic extract of Seed (200 g of powder)	Rats with ischemia-reperfusion (I/R) injury (black cumin post-conditioning)	↑LVDP, RPP, and maximum up/down rate of the left ventricular pressure; protection against oxidative stress (↑SOD and CAT activities; ↓MDA and 4-HNE levels during early reperfusion)	[143]
Ethanolic seed extract (800 mg/kg)	Isoproterenol-induced myocardial infarction in albino rats	↑MI associated alteration and cardiac biomarkers, antioxidant markers, and biochemical activity of cardiac tissue	[144]
TQ (10 and 20 mg/kg, b/w, p.o. for 14 days)	Doxorubicin-induced cardiotoxicity in mice	↓Serum marker and ↑antioxidant enzymes; ↑heart antioxidant defense mechanisms; ↓LPO levels; ↓IL2 level	[148]
TQ (2.5, 5 and 10 mg/ kg, for 28 days)	Diazinon-induced cardiotoxicity in Wistar rats	Act as a natural antioxidant, lessen DZN cardio-toxicity and ameliorated cholinesterase activity	[149]
Hydroalcoholic seed extract (600 mg/kg) and TQ (40 mg/kg)	Angiotensin II-induced hypertension in rats	↓ SBP, MAP, and HR	[145]
Seed extract (300 mg twice daily for 28 days)	Randomized controlled clinical trial in elderly patients with hypertension	A slight but insignificant reduction of blood pressure	[146]
Black cumin virgin oil (twice a day in a dose of 0.5 mL p.o. for 45 days)	Clinical study on patients with mild-moderate hypertension	↓Total cholesterol, LDL and TGs; ↑HDL; ↓systolic pressure and diastolic pressure	[147]

### 6.9. Hepatoprotective Effects

Liver complications account for approximately two million cases of morbidity and mortalities worldwide every year [150]. Multiple etiological factors, such as hepatitis, steatosis, toxic chemicals, radiations, and drugs contribute to liver function impairment [151]. Recent reviews highlighted the hepatoprotective role of black cumin and its bioactive compounds, such as TQ, thymol, and α-hederin [152,153]. Black cumin and its constituents render hepatoprotective action by a range of mechanisms that involve inhibition of lipid peroxidation and oxidative stress, an increase in antioxidant enzymes and total thiol and GSH level, reduction in fat accumulation, and prevention of inflammation and other histopathological features of the liver (Table 6).

Oxidative stress causing lipid peroxidation is one of the major causes of hepatic disease [152,153]. Studies demonstrate that toxicants, such as carbon tetrachloride, arsenic, and acetaminophen induce cellular oxidative stress leading to lipid peroxidation as evidenced by the production of high MDA level in rat models [154,155,156,157,158]. Lipid peroxidation causes the damage of hepatocytes, which is characterized by histopathological changes of liver and high blood levels of hepatic enzymes (ALT and AST). However, supplementation of black cumin extract or its active components stimulates the cellular antioxidant systems and increase the activities of SOD, CAT, and GPx, and level of GSH, thereby reducing the oxidative stress and ER stress (reduced activity of GRP78, CHOP, ATF6, ATF4, XBP1), lowering lipid peroxidation, and improving the longevity of experimental animals [136,154,155,156,157,158,159,160,161,162,163,164,165,166,167].

Another potential cause of hepatic disease is inflammation due to hepatotropic viral infections, chronic hepatitis, cirrhosis, and carcinogen exposure [153]. Studies showed that toxicants trigger the release of excessive inflammatory markers, such as IL-6, Hs-CRP, TNF-α, TGF-β, and NF-κB, and increase liver apoptosis markers, such as Bax, caspase 12, cytochrome c, caspase 9, and caspase 3 [154,165]. However, like oxidative stress, hepatic inflammation can be reduced with black cumin extract and its compounds. Studies demonstrated that supplementation of black cumin extract or its active components increased ratios of anti-inflammatory (Bcl-2 and IL-10) to proinflammatory factors (TNF-α, TGF-β, IL-1, and IL-6) [158,159,163,165,168].

Nitric oxide (NO) is a potent signaling molecule that plays a significant role in liver function [153,169,170,171]. Being a double-edge sword, the association of NO to the liver has mixed outcomes [153,171,172]. For example, a minimal dose of NO serves to block platelet aggregation and thrombosis, increases blood perfusion, and counteracts toxic oxygen radicals in the liver during various disease conditions [152], whereas the excess and persistent presence of this molecule causes hepatic cancer [171,173]. Thus, the pharmacologic modulation of NO synthesis holds promise in the future treatment of liver diseases [171,174]. Black cumin extract or its active components were shown to reduce NO level by modulating the activity of nitrite reductase [154,164,166], indicating that this natural product can protect against NO-mediated hepatic complications. 

Fatty liver, also known as non-alcoholic fatty liver disease (NAFLD) or non-alcoholic steatohepatitis (NASH), occurs when the excess fat builds up in the liver. This condition increases the risk of liver cirrhosis or HCC [175,176]. Several randomized, double-blind clinical trials demonstrated potential hepatoprotective effects of black cumin extract or other preparations (Table 6). Black cumin seed oil (2.5 mL, every 12 h) given to patients with NAFLD for 3 months attenuated hepatic steatosis and caused a significant reduction in blood triglycerides, LDL-C, and aminotransferases and increase in HDL-C level without affecting blood urea nitrogen, serum creatinine, blood cell count and body mass index as compared to the placebo group [167]. These findings indicate that black cumin seed oil may reverse hepatic injury and protect the liver in NAFLD [167]. However, this study is limited by a brief treatment duration and a lack of investigation into the action mechanisms of the black cumin seed oil, liver biopsy, magnetic resonance techniques, and evaluation of fibrosis and inflammation [167]. Although TQ is the main ingredient that was thought to be attributed to these hepatoprotective effects, a clinical trial with TQ against NAFLD is still lacking.

In another randomized controlled trial with NAFLD patients, black cumin treatment (1 g twice a day for three months) caused a decline in BMI, ALT, AST, and overall fatty liver grading on ultrasound following 12 weeks of treatment compared to the placebo group (Hosseini et al., 2018). NAFLD alleviating potential of black cumin oil has also been studied in a randomized, double-blind, placebo-controlled clinical trial, which demonstrates a reduction in FBS level, lipid profiles (TG, TC, LDL, VLDL), liver enzymes (AST and ALT), hs-CRP inflammatory marker, IL-6, TNF-α, and an increase in the HDL-C levels in the interventional group compared to the placebo group [163]. Together, all these compelling evidences suggest that black cumin preparations and its main active compound TQ are likely to be attractive as a promising therapeutic option against a wide range of liver diseases, including NASH.

**Table 6 nutrients-13-01784-t006:** Comprehensive summary on the hepatoprotective effects of black cumin.

Treatment with Doses	Experimental Model	Major Findings (Including Molecular Changes)	References
NSO (2.5 mL/kg BW)	Ibuprofen-induced hepatotoxicity in Swiss albino mice	↓ALT, AST, and ALP	[162]
NSO (2 mL/kg)	STZ-induced diabetic male Wistar rats	↑CAT and GSH; histopathological picture and hepatic glycogen contents	[136]
NSO (300 mg oil/kg BW)	CCl_4_-induced liver injury in rats	↓ MDA, NO and TNF-α, AST and ALT; ↑unsaturated fatty acids	[154]
TQ (100 mg/kg/day BW)	High-dose atorvastatin-induced hepatotoxicity in male SD rats	↓Serum hepatic enzymes, MDA, protein carbonylation, and caspase 3 activity;↑GSH and CAT; histopathological and ultrastructural changes	[155]
TQ (20 or 40 mg/kg, p.o., daily)	Ethanol-induced injury in C57BL/6 mice; TGF-β-induced injury in hepatic stellate cells	↑PPAR-γ; ↑LKB1 and AMPK phosphorylation; ↑SIRT1	[177]
Hydroethanolic seed extract (100, 200, or 400 mg/kg)	LPS-treated rats	↓MDA, NO and IL-6, AST, ALT and alkaline phosphatase; ↑ thiol content, SOD, CAT, serum protein, albumin	[166]
NSO (2 mg/kg/day)	Irradiation-induced liver damage in rats	↓AST, ALT, MDA, SOD, IL-6, TNF-α, TGF-β; ↑IL-10	[158]
NSO (100 mg/kg)	Gibberellic acid-treated pregnant albino rats	↓ALT, AST, MDA, Bax, Hydropic degeneration, Cellular infiltration, periportalFibrosis; ↑SOD, CAT, GPx and Bcl-2	[165]
TQ (400 mg/kg)	Rat model of NAFLD associated with 6-propyl-2-thiouracil (PTU)-induced hypothyroidism	↑CAT, NO, GSH, SOD; ↓MDA, steatosis score, lobular inflammation, NAFLD Activity Score, alpha-smooth muscle actin, intralobular and portal tract and CD68^+^	[156]
TQ (5 or 20 mg/kg)	Rat model of acetaminophen overdose-induced acute liver injury	↓ALT, AST, MAPK Phosphorylation (JNK, ERK and P^38^ phosphorylation); PI3K/mTOR signaling pathway (PI3K, AKT, mTOR, IL-1B, P70S6K); DNA fragmentation and cellular damage; STAT3 phosphorylation, JNK phosphorylation; ↑GSH, GPx, AMPK and LKB1	[159]
NSO (4 mL/kg 48 h)	Carboplatin-induced liver damage in female Wistar-albino rats	↓Apoptotic index, collagen fiber distribution around the central vein, hepatocyte cords preserved	[160]
Seed extract (100, 200, or 400 mg/kg)	PTU-induced hypothyroid rats	↓MDA, ALK-P, AST and ALT; ↑thiol concentration, CAT and SOD, and body weight gain	[157]
TQ (4.5, 9, and 18 mg/kg)	Morphine-induced liver damage in male mice	↑Liver weight; ↓mean diameter of hepatocyte, central hepatic vein, AST, ALT, and NO	[164]
Seed (1 g twice a day for 3 months)	Randomized, double-blind, placebo-controlled clinical trial with NAFLD patients	↓Bodyweight, normal fatty liver grading	[178]
Seed (5 g, drink as tea for 3 months)	Randomized, double-blind, placebo-controlled clinical trial with NAFLD patients	↓AST, ALT, body mass index, and grade of fatty liver	[161]
NSO (2.5 mL/person, every 12 h for 3 months)	Randomized, double-blind, placebo-controlled clinical trial with NAFLD patients	↓Grade of hepatic steatosis, ALT, AST, TG, LDL-C, and HDL-C	[167]
Seed (2 g/day/person, for 12 weeks)	Randomized, double-blind, placebo-controlled clinical trial with NAFLD patients	↑ Insulin sensitivity check index; ↓serum glucose, serum insulin, insulin resistance, hepatic steatosis percentage	[179]
Seed (2 g/day/person, for 12 weeks)	Randomized, double-blind, placebo-controlled clinical trial with NAFLD patients	↓ hs-CRP and NF-κB, TNF-α, hepatic steatosis and its percentage	[168]
NSO (1 g/person, for 8 weeks)	Randomized, double-blind, placebo-controlled clinical trial with NAFLD patients	↑HDL-C; ↓ FBS, lipid profiles (total cholesterol, triglyceride, VLDL, LDL), Liver enzyme (ALT and AST), inflammatory markers (Hs-CRP and IL-6)	[163]

### 6.10. Pulmonary Protective Effects

Inflammation and oxidative stress play a pivotal role in the pathogenesis of various lung disorders [180] such as asthma, chronic obstructive pulmonary disease (COPD) [181], pulmonary fibrosis (PF), acute lung injury (ALI), and lung cancer [182]. Certain herbal remedies are of significant value in the treatments of pulmonary diseases. Several preclinical and clinical studies have investigated the prevention and management of lung diseases describing multiple pharmacological effects of black cumin seed (NS) and its constituents in animal or cellular models of asthma including bronchodilation, anti-histaminic, anti-inflammatory, anti-leukotrienes, anti-fibrotic, and immunomodulatory effects [183,184]. 

As demonstrated in several preclinical investigations (Table 7), NSO and TQ protected against lung damage induced by various chemicals such as lipopolysaccharide (LPS) [185], bleomycin [186,187,188], cigarette smoke [189], or nicotine [190], and cadmium [191]. In almost every case, there was an increase in inflammation, oxidative stress, and apoptosis that accompanied chemical-induced lung injury or fibrosis. NS extracts, NSO, and TQ individually attenuated lung damage by inhibiting these pathological events [185,186,187,188,189,190,191]. Two cell-based in vitro studies also reported anti-inflammatory and anti-histaminic effects of NS extracts [41,192]. 

NS extract has bronchodilatory and preventive effects on asthmatic patients and also on lung disorders in clinical trials [184]. Improvement of asthma symptoms, lung function, and asthma biomarkers by treatment with NS preparations has been reported in several clinical studies [183]. In a randomized single-blind, placebo-controlled clinical trial, black cumin supplement (NS-1, NS-2 = 1, 2 g/day, respectively, for 3 months) given to partly controlled asthma patients with maintenance inhaled therapy increased peak expiratory flow (PEF) by 5–75% in higher dose NS-2 group. Both doses improved PEF variability, serum IFN-γ, and asthma control test score and decreased fractional exhaled NO and serum IgE [193]. NSO capsules 500 mg (two times daily for 4 weeks) were used as a supplementary treatment in a randomized, double-blind, placebo-controlled trial with asthma patients. Compared to the placebo, asthma symptoms and eosinophilia significantly improved with NSO supplementation [194]. These findings provide evidence for the potential benefits of NSO supplementation in the clinical management of asthma. Future studies on an extended period allocating a large number of patients need to be planned to elucidate the effects of black cumin for the management of pulmonary diseases. 

### 6.11. Gastroprotective Effects

The gastrointestinal system can be affected by several factors, including drug-induced secondary effects (side-effects), microbial infections, and various forms of ulcers. Natural products, including black cumin seed and TQ were found to be effective in protecting against these abnormalities as presented in Table 8.

Clinical application of cisplatin (CP), a widely prescribed drug in cancer chemotherapy, is sometimes associated with undesirable side effects, particularly in the gastrointestinal tract [195]. As a natural pharmacological agent, NSO or TQ show protection against CP-induced gastrointestinal damage. For example, NSO administration (2 mL/kg body weight, p.o.) can help to prevent the accompanying gastrointestinal dysfunction in cisplatin (CP) chemotherapy [196]. TQ has shown protective effects by a considerable reduction of specific activities of brush border membrane (BBM) marker enzymes, restoring the redox and metabolic status of intestinal mucosal tissue and preserving intestinal histoarchitecture against CP-induced gastrointestinal damage in rats [197]. Another observation reported similar results showing that both NSO and TQ administration can ameliorate CP-induced alterations on the enzymatic and non-enzymatic parameters of the antioxidant defense system in the intestinal mucosa [198]. Notably, NSO appeared to be more effective than its major constituent TQ in protecting against CP-induced gastrointestinal dysfunction [198]. 

Gastric ulcer is a multi-step disease and healing requires a complex process including repair and re-architecture of the gastric mucosa. In a rat model of acetic acid-induced gastric ulcers, rhamnogalacturonan-I type pectic polysaccharide of black cumin can mediate ulcer healing by modulating signaling pathways involved in either ulcer aggravation or healing process [199]. The potentiality of black cumin in preventing or curing gastric ulcers has also been reviewed elsewhere [200]. TQ also reduced the volume and total acidity of gastric secretion in comparison with carbachol [201], suggesting that TQ can be efficiently used in peptic ulcer treatment and other conditions, such as dyspepsia, gastritis and reflux esophagitis, which are results of hyper gastric acidity [201].

A major complication following gastrointestinal surgery is an anastomotic leak that can be treated with black cumin. In a rat model of colonic anastomosis, black cumin protected against ischemia/reperfusion injury by improving tissue and serum levels of total oxidant and antioxidant status, total thiol, and hydroxyproline, and by reducing interleukin-6 and TNF-alpha [202]. In addition, the histological tissue damage was milder after black cumin treatment [202].

A clinical study showed the effectiveness of a combination of black cumin and honey (Dosin) in the eradication of gastric *H. pylori* infection. For this purpose, nineteen patients with *H. pylori* infection without a history of peptic ulcer, gastric cancer, or gastrointestinal bleeding, were suggested to receive one teaspoon of Dosin (6 g/day of black cumin as ground seeds and 12 g/day of honey), three times a day after meals for two weeks. The major findings included a negative urea breath test (UBT) and a notable decrease in the median and interquartile range (IQR) of total dyspepsia symptoms without adverse events, concluding that Dosin can be a potential anti-*H. pylori* and an anti-dyspeptic agent [203]. However, in another clinical study allocating a total of 46 patients with active mild to moderate ulcerative colitis, supplementation with 2 g of black cumin powder for 6 weeks showed no remarkable difference in serum total antioxidant capacity and nuclear factor-kB levels with a reduction in stool frequency [204]. Additionally, some adverse effects, including nausea, bloating, and a burning sensation, have been reported after NSO administration in functional dyspeptic patients, and there was a slight increase in liver and kidney enzymatic markers following the consumption of NSO and crushed seeds [205]. Considering the findings of preclinical and clinical studies, it is suggested that black cumin and TQ may be potential gastroprotective agents. However, further clinical trials of black cumin administration with a different pattern are, therefore, necessary to suggest clinical use.

### 6.12. Effects on Fertility and Reproduction

Reproductive soundness is determined by the testosterone level, sperm count and characteristics, and semen quality in males and the estrogen, progesterone, and other reproductive hormone levels, and ovarian function in females. Black cumin seed, particularly its oil (NSO) and the main constituent thymoquinone, positively influence these fertility indicators and, thereby, improve reproductive performance as reported in numerous preclinical and clinical studies. Oral gavage of NSO for 45 days significantly improved testicular spermatogenesis, semen parameters, and seminal vesicle development in rats [206]. In the same study, when rats were co-administrated with acetamiprid (a commonly used neonicotinoid insecticide), NSO was found to ameliorate acetamiprid-mediated toxic effects on the reproduction, including altered testicular weight gain, semen quality, and serum testosterone levels [206]. At the molecular level, the decreased serum testosterone levels induced by acetamiprid inhibits steroidogenic activities, since NSO increased testosterone (testosterone levels in acetamiprid-exposed rats was one-fourth than that of co-administered with NSO), thus positively influence the functions of the testis and spermatozoa [206]. Besides, reproductive organs/tissues (e.g., testis, and testicular cells) are highly sensitive to oxidative stress [207,208]. As such, NSO might directly influence antioxidant defense machinery of reproductive cells/tissues by enhancing the activities of antioxidant enzymes (e.g., superoxide dismutase, glutathione peroxidase, and catalase) [206]. These potential antioxidant activities in returns downregulate the ROS generation and lipid peroxidation in reproductive cells/tissues, and subsequently promote fertility and reproduction [206,209]. The similar beneficial effects of NSO on male fertility and reproduction have been supported by several other contemporaries using rat and mice models [210,211,212]. In contrast to the aforementioned preclinical evidence, so far there is only one clinical trial that has confirmed the effects of black cumin on male reproduction. The study demonstrates that NSO (2.5 mL twice daily p.o. for 60 days) significantly improved semen quality (e.g., volume and pH) and functional parameters of spermatozoa (e.g., sperm concentration, motility, and morphology) in the Iranian infertile men compared to those in the placebo control group (liquid paraffin) [213].

Consistent with the effects of black cumin on male fertility and/or reproduction, several toxicopharmacological/clinical studies have also reported its potential role in female reproduction. In an observational study, Latiff and colleagues investigated that oral administration of black cumin powder (1600 mg/day) for 12 weeks in perimenopausal Iranian women significantly reduced the prevalence and severity of the menopausal symptoms [214]. Besides, women exposed to black cumin powder significantly improve perimenopausal weight gain, and the circulating lipid, glucose, and hormonal levels and subsequently affect their reproductive health [214]. In a randomized double-blind clinical trial, [215] demonstrated that women who received black cumin tablet (500 mg) alongside mefenamic acid (250 mg, NSAID drugs commonly used to treat mild to moderate pain) experienced significantly lower postpartum pain within 2 h of delivery than those who received only mefenamic acid [215]. Apart from the clinical observations, in vitro supplementation of thymoquinone (1 mM/mL) to mouse ovarian granulosa cell modulates overall NF-κB signaling cascades (major transcription factors in the regulation of the inflammatory process, including ovulation) by downregulating cyclooxygenase-2 expression and ROS levels [216]. In the same study, when the polycystic ovary rat was injected with thymoquinone (0.75 mg/100 mL) twice, a significant recovery of ovarian function (including reduction of ovarian cyst formation, increased rate of ovulation, and associated ovarian function) was observed [216]. Several reproductive hormones, including estrogen, luteinizing hormone, thyroxine, triiodothyronine, and thyroid-stimulating hormone and overall reproductive performance were also improved in female rats following NSO treatment (1 mL/kg body weight/day) for 30 consecutive days [217]. NSO (1% or 2%) given to female mice improved overall health condition and reproductive performance by modifying physiological indices and oocyte quality [218]. Based on the aforementioned observations, it is decisively proved that black cumin possesses an extraordinary role in the improvement of fertility and reproduction (overall effects are compiled in Figure 5). However, more studies should be conducted to determine the relative safety and efficacy of black cumin for its possible clinical application.

### 6.13. Protection against Skin Diseases

Skin diseases are characterized by papules or pustules or comedones caused by viruses, bacteria, fungi, and parasites. Skin diseases are of various kinds, including skin rashes, acne, vitiligo, eczema, psoriasis, and deep wound. Several studies demonstrated the efficacy of black cumin seed for the treatment of various skin diseases (Table 9).

#### 6.13.1. Wound Healing

The hydroethanolic extract of black cumin healed 20% to 40% of skin wounds in diabetic rats compared to control rats [131]. The anti-inflammatory and antimicrobial properties of black cumin might be involved in this wound healing [131]. In diabetic wounded rats, black cumin essential oil (NSO) improved the antioxidant status associating wound healing process by increasing activities of GPx, SOD, and CAT and GSH level [219]. A mixture of cold-pressed NSO and honey also reduced wound surface area significantly in Wistar rats [220]. Treatment with a combination of 50% NSO cream and 50% perforatum (HP) oil was shown to heal skin wounds [221]. Bannai et al. reported that hot methanol extract (70%) of black cumin seed healed infected wounds in rabbits by enhancing scab formation and granulation tissue formation in wound areas [222]. Film-forming polymeric solution containing black cumin extract can be used as an alternative antibacterial product for the treatment of skin infection caused by the two pathogens: *S. aureus* and *S. epidermidis* [223]. Future studies should focus to disentangle the mechanism of black cumin mediated wound healing. It is, however, predicted that anti-inflammatory, antioxidant and antimicrobial properties might function to heal burn wounds.

#### 6.13.2. Acne Vulgaris

Acne vulgaris is one of the dermatological conditions that occur because of the obstruction and chronic inflammation of sebaceous follicles in the skin. Several studies suggest acne vulgaris ameliorating properties of black cumin seeds (Table 9). Nawarathne et al. showed that the application of three topical gels formulated with ethyl acetate extract of black cumin seeds for 30 days inhibited growth of *S. aureus* and *P. acnes* [224]. Application of TQ-loaded ethosomes gel formulation (THQ–EGF) has been shown to decline the number and size of sebaceous glands in rats suggesting TQ as an effective treatment option for acne vulgaris [225]. Recently, a randomized double-blind controlled clinical trial with 60 patients who received hydrogel made of black cumin twice a day showed a significant reduction in the number of comedones, papules, and pustules without any side effects, suggesting that black cumin may reduce the symptoms of acne vulgaris [226].

**Table 9 nutrients-13-01784-t009:** Comprehensive summary on the skin protective effects of black cumin.

Treatment with Doses	Experimental Model	Major Findings (Including Molecular Changes)	References
Hydroethanolic *seed* extracts (20% or 40%)	Diabetic skin wounded male Wistar rats	↑ Anti-inflammatory and antimicrobial effect	[131]
NSO-containing cold cream	Wounded (dermis + epidermis) male albino Wistar rats	↑Epithelialization rate	[227]
A mixture of 1:1 ratio of honey and cold-pressed NSO	A circular excision wound in the back region of male albino Wistar rats	↓ Wound surface area	[220]
Cold pressed NSO (3 mL)	Double-blind randomized study with Albino rabbits	↑ Wound contraction; ↓inflammation	[228]
NSO cream (50%)	Female Wistar-albino rats with 84 excisional skin wounds on the backs	↑ Epithelialization and granulation	[221]
70% hot methanolic seed extract	Male adult rabbits with skin incision	↑Wound healing without any infection	[222]
Ethanolic seed extract (IC50 values of 71.54 ± 3.22 μg/mL)	Murine macrophage leukemia cell line (RAW 264.7), human promyelocytic leukemia cell line (HL-60), Murine embryonic fibroblast cell line (3T3-CCL92)	↑Wound closure	[229]
Hydrodistillation of seed [0.6% (w/w) of essential oil] (10 μL)	Diabetic Sprague–Dawley male rats with 2 excision wounds on the upper back of each animal with a dermal punch	↓ Oxidative stress and lipid peroxidation	[219]
NSO (2 g/kg b.w.)	Albino Wister male rats	↑Formation of wound collagen	[230]
Ethyl acetate seed extract (15%)	Gel prepared with the seed extract	↓Growth of *S. aureus* and *P. acnes*	[224]
TQ	Ethosomes gel with TQ	↓Number and size of sebaceous glands	[225]
Pure NSO (5 mg/kg b.w.)	Male Albino Rats having psoriasis-like skin inflammation	↓ IMQ-induced psoriasis-like inflammation	[231]
Hydrogel of hydro-ethanol seed extract	Randomized double-blind controlled clinical trial with mild-to-moderate acne vulgaris patients	↓Number of comedones, papules, and pustules;no side effect	[226]
NSO-containing cream	Vitiligo patients, 47 body surface areas were affected	↑Repigmentation	[232]
NSO	Three patients with contact dermatitis	Controlled contact dermatitis	[233]

#### 6.13.3. Vitiligo

Vitiligo is a long-term skin disorder caused by a speckle of skin depigmentation. A clinical study with 33 vitiligo patients aimed to understand repigmentation rate in different body surfaces upon application of black cumin seed oil. The study showed that the highest repigmentation rate was achieved in hands, face, and genital region after applying black cumin seed oil-containing cream in the morning and evening [232].

### 6.14. Bone Regenerative Effects

Black cumin seeds have significant positive effects on bone formation and healing as demonstrated in the experimental osteoporotic models [234]. Black cumin has shown bone regenerative potential that involves bone marrow with dilated blood vessels during active bone formation, dense trabeculae and large aggregate osteocytes in the tooth extraction socket healing process in rabbits [235]. Black cumin seed extracts exhibited a significant effect in calvarial defected ovariectomy-induced osteoporosis in rat models by promoting osteoblast proliferation, ossification and decreasing osteoclasts [234]. Oral administration of TQ (10 mg/kg) was shown to ameliorate bone defect in the rat model. However, different doses and local applications are recommended for further studies of TQ on bone healing [236]. NSO incubated dental pulp mesenchymal stem cells (DP-MSCs) isolated from third molars of human patients (15–20 years of age, *n* = 5) presented clear and compact calcium granules, indicating potential osteogenic differentiation in DP-MSCs [237].

In a clinical trial on 12 healthy patients randomly divided into two groups, including a test group (six patients NS treatment) and a control group (six patients without NS treatment), NS gel improved bone quality thereby increasing bone density and improved peri-implant tissues in the test group after 3 and 6 months [238]. Apart from this clinical trial, no other significant human trials have been performed to assess the bone regeneration capacity of NS seeds. Further clinical studies are, therefore, required to translate preclinical findings (Table 10) into clinical use.

### 6.15. Nephroprotective Effects

Globally there are about 13.3 million cases of acute kidney injury (AKI) with an estimated annual incidence of 11.3 million in developing world annually. Patients with AKI have an increased risk of developing chronic kidney diseases (CKD) and vice-versa [239]. The therapeutic management of kidney diseases encompasses pharmacological targeting of pathological events that contribute to the disease progression [240]. Black cumin seeds, NSO and TQ, comparable to some other natural products, have shown therapeutic promises against kidney diseases.

A growing body of literature suggests protective effects of black cumin, particularly NSO against a range of chemical/drug/heavy metals/pesticides-induced nephrotoxicity. For instance, black cumin ameliorated the damaging effects of commonly prescribed cancer chemotherapeutic agents such as methotrexate [241,242] and cisplatin [243,244,245]. These protective effects stem from the prevention of lipid peroxidation and enhancement of antioxidant enzyme activity in renal tissues of chemotherapy-treated animals [246]. Moreover, in different rat models, black cumin protected against nephrotoxicity induced by several anti-inflammatory agents such as acetylsalicylic acid [247], aspirin [248], and paracetamol [249]. On the other hand, NSO prevented haloperidol (an antipsychotic drug) induced nephrotoxicity in rats [250]. Moreover, NSO functions as an antidote against nephrotoxicity and renal damage caused by high doses of penconazole, a triazole fungicide widely used in agriculture, human and veterinary medicine [251]. 

Treatment of TQ inhibited arsenic [252] and cadmium [253], and sodium nitrite [254]-induced nephrotoxicity by reducing oxidative damage, apoptosis, and inflammation. In addition, oral administration of NSO protected kidney tissues against sodium nitrite [255]- and CCl_4_-induced oxidative stress and inflammation [154,256]. In addition, pretreatment with black cumin has a protective effect against reperfusion-induced renal damage by inhibiting apoptosis and cell proliferation [257]. Black cumin can also ameliorate nephrolithiasis and renal damages [258], and unilateral ureteral obstruction (UUO)-induced kidney damage [259]. In addition, complementary uptake of black seeds or NSO preparation benefits CKD patients [140,260,261]. Together, it can be anticipated that black cumin and TQ may be useful in protecting against a wide range of kidney complications.

### 6.16. Anti-Arthritis Effects

Arthritis is an inflammatory condition of joints, generally affecting the elderly. Rheumatoid arthritis is a chronic inflammatory autoimmune disease that is characterized by synovial membrane inflammation, swelling, autoantibody production. Due to potent anti-inflammatory action, *Nigella sativa* oil (NSO) and thymoquinone (TQ) were shown to be effective against arthritis, including RA. As some reviews have already highlighted anti-arthritic potential of NSO and TQ elsewhere [262,263], we better addressed some recent progress. For example, NSO (1.82 mL/kg) reduced proinflammatory mediators (such as IL-6) and improved (56%) adjuvant induced-arthritis in the rat model [264]. Black cumin extract was supplemented in collagen-induced arthritis mice and exhibited a significant decrease in the inflammation score and neutrophil infiltration, exhibiting its anti-arthritic property [265]. The anti-arthritic effect of TQ (10 mg/kg in 0.1% DMSO) was observed in a rat model of arthritis with a marked reduction in the mRNA levels of toll-like receptor 2 (TLR2), TLR4, IL-1, NF-κB, and TNF-α [266]. Intra-articular injections of 0.3 mL of NSO (5 weeks) in 5–7 months old male New Zealand white rabbits prevented the degeneration of cartilage at the prior stages of osteoarthritis [267]. A double-blind clinical trial with topical application of NSO observed a significant reduction in osteoarthritis pain in elderly people [268]. Another randomized double-blind, placebo-controlled clinical trial investigated the effect of black cumin seed powder (2 g/day) in knee osteoarthritis patients (40–70 years) with significant improvement in the active intervention group without any adverse effects [269]. NSO in a dessert spoon amount, thrice a week for one month alleviated the symptoms of knee osteoarthritis in geriatric individuals [270]. All this evidence provides significant information supporting its anti-inflammatory and immunomodulatory effects.

### 6.17. Protection against Emerging Diseases

Several in silico studies have reported pharmacological potentials of black cumin and its various compounds against severe acute respiratory syndrome coronavirus-2 (SARS-CoV-2), the causal pathogen of recently emerged coronavirus diseases 19 (COVID-19) pandemic [271,272,273]. A line of recent reviews also revisited pharmacological potentials of black cumin in the prevention of COVID-19 and the associated complications [274,275,276,277].

A 46-year-old HIV patient treated with black cumin concoction was completely recovered and remained seronegative for a period of six months [278]. In another case report by the same group, HIV infection in a 27 year old woman completely seroreverted following black cumin and honey therapy (60:40, 10 mL thrice daily for a year) and three children born after the woman was HIV positive remained uninfected [279]. Although these findings are promising, these studies are limited by lower sample size and lack of replicability. Future studies with similar approaches and larger sample sizes are necessary to determine the antiviral efficacy of black cumin seed against HIV. 

### 6.18. Black Cumin and TQ as a Promising Antidote

In the current age of industrialization with modern lifestyle, the toxicity caused by toxic chemicals, especially mercury, arsenic, iron, lead, chromium, malathion, and cadmium has become an important public health concern worldwide. Many of these toxicants present in daily used utensils and commodities end up in foods and, thus, in the human body. Toxic metals accumulate in the tissues of various organs as they are not metabolized and thereby interact with the vital cations, inhibiting enzymes and triggering oxidative stress, which are responsible for inducing various diseases such as neurotoxicity, diabetes, cancer, cardiovascular diseases, infertility, and risk of renal damage [280,281]. There are two main modalities for managing these toxicities: chelation therapy to chelate toxic metals and antioxidant therapy to counteract oxidative stress. While long-term use of artificial chelating agents and antioxidants usually has adverse consequences, natural products such as black cumin could be a better option as they have no or minimal adverse effects [281].

In the individual sections, system/organ/disease-specific antidotal effects of black cumin and TQ against natural and chemical-induced toxicities and the mode of actions have already been discussed. In this section, we only reviewed the literature that reported generalized findings on the antidotal and protective effects of black cumin and its constituents and are not discussed yet (Table 11). Hassan reported that black cumin extract (25–200 μg/mL) chelated Fe (II) and scavenged hydroxyl radical (OH^−^) [282]. TQ was shown to attenuate toxicity by the formation of a complex with Cr(VI) or by conversion of Cr(VI) to Cr(III) [283]. In another study with endothelial cells of human umbilical vein, TQ reduced 2-tert-Butyl-4- hydroquinone-induced cytotoxicity by anti-apoptotic and chromosome protecting effects. In several in vivo studies, TQ has shown its antioxidant-dependent antidotal effects against toxicities in rodents induced by malathion and fipronil [284,285] and diazinon [286]. TQ also reduced venom-induced acute toxic shock in male rats by antioxidant and antiallergic functions. Therefore, black cumin and its constituents, particularly TQ, could be a promising natural antidote due to its chelating ability, anti-allergic, anti-clastogenic, antioxidant and anti-apoptotic effects [287].

### 6.19. Black Cumin as a Galactagogue

Black cumin is traditionally used by the lactating mother due to its promotional effect on milk production. Experimental evidence also endorses black cumin as a natural galactagogue [289,290]. Hosseinzadeh and colleagues demonstrated that black cumin in the form of aqueous (0.5 g/kg) and ethanolic extracts (1 g/kg) augmented milk production with a yielding of about 31.3% and 37.6% more milk than control, respectively [291]. It has been suggested that the estrogenic constituents of black cumin, such as anethole, might contribute to its lactogenic activity as this dopamine analog stimulates prolactin release and increase milk production by antagonizing at the dopamine receptor site [291]. However, histoarchitectural evaluation of mammary glands and profiling of pituitary prolactin are warranted to clarify the detailed mechanism of the lactogenic activity of this plant. 

## 7. Molecular Mechanisms Underlying the Pharmacological Effects across Health and Disease Conditions

Black cumin and TQ were shown to exert diverse pharmacological and health effects through modulating multiple cellular signaling systems. The notable molecular pathways targeted by black cumin and TQ are Nrf2, NF-κB, TLR, SIRT1, AMPK-SIRT1-PGC-1α, PPAR, and PI3K/Akt signaling, which are shared across health/disease conditions (Figure 6).

Antioxidant activity of black cumin and TQ is amongst the pharmacological effects that underlie many of its health benefits and has been manifested by their capacity to enhance expression of enzymatic (such as SOD, GPx, CAT, and HO-1) and non-enzymatic (such as GSH) antioxidants, lowering various oxidative markers (such as ROS, MDA, LPO, and 4-HNE). The genetic expression of these antioxidants molecules is under the transcriptional regulation of Nrf2.

Activation of Nrf2 by either cellular redox status or pharmacological intervention leads to the up-regulation of over 250 genes encoding proteins that are involved in antioxidant defense systems, redox homeostasis, and xenobiotic detoxification [81]. Increased expression of antioxidant molecules and subsequent decline in oxidative markers by black cumin and TQ in various pharmacological effects indicate the involvement of Nrf2 activation [51,55,58,69,132,136].

In addition to its potentials on activating cellular antioxidant defense system, black cumin can directly scavenge free radicals, as demonstrated in several in vitro chemical assays like DPPH assay [35,292,293,294]. However, by virtue, TQ has a relatively poor capacity to quench free radicals because of its oxidized form [295]. This observation strengthens the idea that TQ can exert its antioxidant capacity by activating the Nrf2-dependent antioxidant defense system. However, thymohydroquinone, the reduced form of TQ, possesses a high radical-scavenging capacity [295]. It has been speculated that the conversion of TQ to thymohydroquinone can occur in cells and that the electron transport chain may have an important role in the antioxidant action of TQ.

While toll-like receptors (TLRs) signaling ensures protective immune response by recognizing invading pathogens and tissue-derived endogenous molecules, its overactivation perturbs the immune homeostasis by sustained release of pro-inflammatory mediators and subsequently underlies the development of many inflammatory diseases [296]. TQ may improve inflammatory response in Alzheimer’s disease model by downregulating the expression of TLRs signaling components as well as their downstream effectors NF-κB and IRF-3 [54].

Modulatory role of TQ in autophagy-an evolutionarily conserved cellular process that recycles defective and unwanted cell components and invading pathogens to retain cellular homeostasis has also been documented [297]. Protection against neuroinflammation by TQ in LPS-activated BV-2 microglia involved autophagy induction through activation and nuclear accumulation of SIRT1 [50]. Mitophagy, a form of autophagy that clears defective mitochondria, is regulated by parkin and Drp1 expression. An alteration of parkin and Drp1 expression may lead to impairment of mitophagy triggering apoptosis and neurodegeneration in the brain. Rotenone hindered parkin-mediated autophagy by upregulating Drp1 expression, which was ameliorated by TQ treatment [59].

The anticancer potentials of black cumin and TQ are vested on their capacity to regulate various cellular pathways that are implicated in proliferation, apoptosis, cell cycle regulation, carcinogenesis, angiogenesis, and metastasis [121]. Most of the anticancer actions of black cumin and TQ are reported to mediate by regulating cellular redox systems [117] through which both TQ and black cumin can inhibit cell proliferation, migration/invasion, and tumor growth by directly acting on growth factor signaling systems, such as EGFR/ERK1/2, Akt/mTOR/S6, Wnt, β-catenin, and VEGF signaling [121,298,299]. TQ can prevent cancer development by its antioxidant function and can hinder cancer progression through its pro-oxidant function [121]. Besides, TQ enhanced chemosensitivity to chemotherapeutics and chemopreventive molecules by downregulating inflammatory signaling pathways and enhancing tumor-suppressing genes [116,118,120,300].

As a master upstream kinase, LKB1 phosphorylates and activates AMPK and many other kinases that play a fundamental role in the regulation of cell growth and metabolism [301]. The LKB1–AMPK pathway acts as a cell metabolic checkpoint, arresting cell growth under low intracellular ATP conditions, such as in nutrient-deficient states [301]. Energy overload may suppress LKB1–AMPK signaling, leading to increased cancer risk in patients with obesity or diabetes. Whereas, activation of LKB1–AMPK signaling might contribute to the suppression of cancer risk and, thus, pharmacological modulators, such as TQ, which was shown to activate LKB1–AMPK signaling [106], could have therapeutic promise in cancer prevention.

Apart from the aforementioned mechanism, there still remain other (albeit not less significant) signaling systems that are targeted by black cumin and TQ, such as unfolded protein response (UPR). Triggering of endoplasmic reticulum (ER) stress is a common phenomenon in several pathological conditions such as hypoxia/reoxygenation and oxidative stress. ER homeostasis is crucial for proteostasis and its disruption results in the buildup of unfolded and misfolded proteins in the ER lumen. Consequently, UPR is activated to resolve this protein-folding defect and thus to restore ER homeostasis. In the case of an insufficient UPR, pharmacological activation can play a therapeutic role in mitigating ER stress. Attenuation of ER stress by TQ suggests its protective role in maintaining proteostasis. Moreover, black cumin nanoemulsion promoted Aβ clearance, thus maintained protein homeostasis in the brain, by upregulating LRP1 [52], a type I transmembrane glycoprotein expressed abundantly in neurons that facilitates trafficking and degradation of Aβ [302].

## 8. Drug Interaction and Nanoparticle-Mediated Drug Delivery

Drug interactions with TQ were investigated using TQ or black cumin extract in dog, rat, or rabbit model (Table 12). Mostly 5 mg/kg PO or 10 mg/kg IV dosage of TQ was used. TQ reduced the bioavailability of CsA [303], sildenafil [304], and phenytoin [305] while increasing the bioavailability of GBC [306] and amoxicillin [307]. Synergistic effects on blood glucose levels were reported with TQ and GBC [306]. Co-administration of TQ does not alter the pharmacokinetic parameters of theophylline [308] and CBZ [309].

Significant improvement has been achieved in nanoparticle-based drug delivery strategies of TQ indicated for various diseases as presented here (Table 13) and elsewhere [310,311]. Apart from improved TQ delivery through the oral route, further development should, therefore, focus on reduced hydrophobicity, thermal instability, and slower degradation of TQ.

**Table 12 nutrients-13-01784-t012:** Drug interaction with black cumin and corresponding pharmacological effects.

Active Compound/ Extract	Interacting Drug	Experimental Model	Route of Drug Administration	Dose of Drug	Effects on Pharmacological Parameters	Reference
TQ	Cyclosporine A (CsA)	Rodents	PO * IP	TQ (PO, 10 mg/kg) CsA: 10 mg/KG (PO and IP)	Bioavailability: oral CsA reduced by 32% but IP CsA was not affected Chronic CsA effect (Increase fasting glucose, cysteine C and marked kidney alteration) was reversed by TQ	[303]
TQ	Glibenclamide (GBC)	Rat	PO	10 mg/kg	Plasma concentration of GBC increased by 13.4% (Single dose) and 21.8% (multiple doses) with TQ Synergistic effect on glucose level	[306]
TQ	Quercetin (QR)	Fluorescence- assays	TQ (purity ≥95%, HPLC	Assay	An insignificant inhibitory effect on the activity of CYP1A2 or CYP2E1. Moderate to a strong inhibitory effect on CYP3A4 activity. Moderate inhibitors of the CYP2C9. QR has a moderate inhibitory effect on CYP2C19 and a strong inhibitory effect on CYP2D6.	[312]
Black cumin	Amoxicillin	Rat Model	PO	25 mg/kg BW	Methanol and hexane extracts increased the permeation of amoxicillin significantly; Enhanced amoxicillin availability in both in vivo and in vitro	[307]
Black cumin	Amoxicillin	Rat sac model	Rat sac	Seed extract	The methanolic extract improved intestinal permeability of amoxicillin in the in vitro experiments in a dose-dependent manner	[313]
Black cumin	Sildenafil	Beagle dogs	PO	Sildenafil 100 mg	Reduced AUC0-∞, C max and t 1/2 of Sildenafil	[304]
Black cumin	Cyclosporine (CCS)	Rabbit	PO	200 mg/kgCCS (30 mg/kg)	Co-administration significantly decreased the C(max)-35.5% and AUC (0-∞)-55.9%	[314]
Black cumin	Phenytoin	Beagle dogs	PO	Phenytoin 50 mg	Drastic reduction of elimination and to a lesser extent on VoD at steady state (Vss) with a consequent reduction of area under the curve (AUC0-∞) by about 87%	[305]
Black cumin	Theophylline	Beagle dogs	PO	200 mg	No significant effect on theophylline disposition as measured by *C*_max_, *T*_max_, AUC_0–∞_, and CL/*F*	[308]
Black cumin	Carbamazepine (CBZ)	Rabbit	PO	Black cumin (200 mg/kg) or Lepidium sativum (150 mg/kg)	Concurrent use of Lepidium sativum but not black cumin alters the pharmacokinetics of CBZ	[309]
Black cumin	CYP2C11	Wistar rats	PO	300 mg/kg	Significantly inhibited the mRNA and protein expression levels of CYP2C11 in a dose-dependent manner.	[315]

* PO—per os, IV: intravenous, IP: intraperitoneal.

**Table 13 nutrients-13-01784-t013:** Recent progress in nanoparticle-mediated drug delivery of TQ.

Nanoparticles	Method of Preparation	Size	Zeta Potential	Dose (EC)	Experimental Model	Indication	Benefits/ Advantages	References
TQ-loaded nanocapsule	Nanoprecipitation	70.21 nm	–45.3 mV	10 mg/kg TQ-loaded NCs	Streptozotocin plus nicotinamide–induced diabetic Wistar female albino rats	Anti-diabetes	Decrease blood glucose and HbA1c improvements in body weight and lipid profile	[316]
TQ-loaded phospholipid nanoconstructs	Micro-emulsification technique	83.44 nm	−0.65 mV	20 mg/kg	Wistar rats	Hepatoprotection	Enhanced hepatoprotective effect ratified by histopathological analysis Reduction in the ALP, ALT, AST, bilirubin, and albumin level	[317]
TQ-capped iron oxide nanoparticles	Co-precipitation method	10 nm	−33.4 ± 1.5 mV	0.05, 0.1, 0.15, 0.2, and 0.25 mg/mL	MDA-MB-231 (epithelial, human breast cancer cell)	Chemo-Photothermal Therapy of Cancer	TQ bioavailability	[318]
TQ-loaded chitosan-lecithin micelles		50 nm to 100 nm	-	200 µL of 20 mg/mL of TQ-PMs	Balb/c mice	Wound healing efficacy	Increased formation of somewhat organized collagen fibers and blood vessels Less inflammation	[319]
TQ-loaded nanoformulation	Emulsion solvent evaporation method	97.36 ± 2.01 nm	−17.98 ± 1.09	10 mg kg^−1^	Albino rats	Epilepsy	Improved THQ-brain- bioavailability improved the seizure threshold treatment	[320]
TQ-loaded Chitosan nanoparticles (TQ-TPP-Cs NPs)	Bio-fabrication and statistical optimization	391.4 ± 78.35	30.9 ± 3.02 mV	(141.91 mg/kg	Wistar rodents	Depression	Efficient in ameliorating the behavioral and neurochemical changes Improved motor activity and swimming time and increased immobility time Serotonin, norepinephrine and dopamine	[321]
TQ-loaded, hyaluronic acid (HA)-conjugated Pluronic^®^ P123 and F127 copolymer nanoparticles (HA-TQ-Nps)		15–20 nm	--	1.5, 2, 3 μg/mL	Two human TNBC cell lines such as MDA-MB-231 and MDA-MB-468	Triple-negative breast cancer	Pro-apoptotic, anti-metastatic and anti-angiogenic activity Retarded cell migration of TNBC cells through up-regulation of microRNA-361 which in turn down-regulated Rac1 and RhoA mediated cell migration and also perturbed the cancer cell migration under the influence of the autocrine effect of VEGF-A	[322]
TQ-loaded polymeric nanocapsules	nanoprecipitation technique	217 to 231.5 nm	−36 to −39 mV	100, 200, 300, 400 μM	Colon cancer cell lines (HT-29, HCT-116, Caco-2)	Colon cancer	Higher cytotoxicity against HT-29 cells overexpressing sigma receptors Delineating anisamide as a promising ligand for active colon cancer targeting.	[323]
TQ solid lipid nanoparticles	solvent injection methods			20 mg/kg	Male Albino Wistar rats	Depression	Reduced pro-inflammatory cytokines (IL-6 and TNF-α) Reduced Activation of Indoleamine-2,3-dioxygenase Increased BDNF	[64]
Ethosomic TQ	Conventional method	20 ± 1 nm	−63 ± 2 mv		Human epithelial breast cancer cell lines MCF-7	Breast cancer	Highly cytotoxic for cancer cells	[324]

## 9. Safety Evaluation of Black Cumin-Based Therapeutics

Several studies have been dedicated to evaluate toxicological effects of black cumin and TQ [325,326,327,328]. Moreover, along with the biological effects, the toxicity of black cumin and TQ has also been assessed in some studies [329]. Very recently, Mashayekhi-Sardoo and colleagues also reviewed the in vivo toxicological profile of TQ [330]. The pattern and degree of toxicity of TQ depend on the dose size, route of administration, duration of exposure, and model animal. In general, non-toxic and toxic doses of TQ administered intraperitoneally were much lower than those of the oral route (Figure 7a,b).

**Figure 7 nutrients-13-01784-f007:**
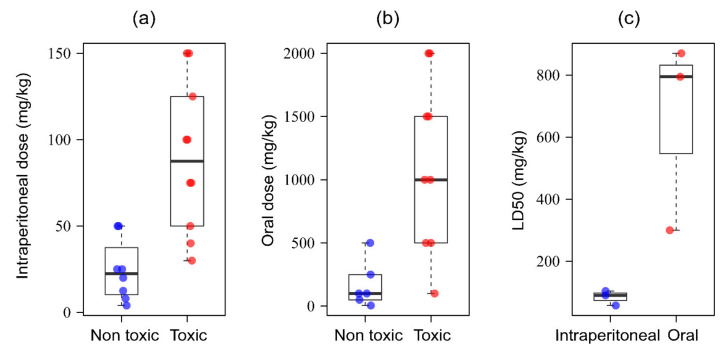
Therapeutic window along with the toxic and lethal dose limit of intraperitoneally and orally administered TQ. (**a**) Toxic (*n* = 10) and non-toxic doses (*n* = 8) of TQ when intraperitoneally administered in animal models. (**b**) Toxic (*n* = 6) and non-toxic doses (*n* = 9) of TQ when orally administered in animal models. (**c**) LD50 values of TQ as determined after intraperitoneal (*n* = 3) and oral (*n* = 3) administration in animal models. *n*, number of test doses retrieved from ‘n’ number of studies [129,327,328,329,331,332,333,334,335,336]. Bold lines indicate medians. Boxes enclose 25th to 75th percentiles. Error bars enclose the data range, excluding outliers. Dots are data points of each tested dose; dots that are vertically outside the error bars are outliers, >1.5 times the interquartile range.

To evaluate the acute toxicity, mice administered orally with TQ (at 2 and 3 g/kg, dissolved in corn oil) exhibited hypoactivity and affliction in breathing before death within the first 3 h after administration along with a marked decline in the weight of vital organs (liver, kidneys, and heart) and liver and kidney GSH content in the surviving animals. Mice were quite normal below these toxic doses [328]. In another study investigating the effects of TQ on blood lipid profiles, low doses (less than 6 mg/kg/day for 1–14 day long timeframe) were tolerable as evidenced by no signs of toxicity; but, at the high dose (8 mg/kg/day), most of the animals succumbed by the end of the first week of treatment and others that survived presented signs of peritonitis [329].

Jrah Harzallah et al. reported that TQ at 80 mg/kg, but not ≤40 mg/kg, markedly caused chromosomal aberrations and DNA damage in the liver and kidney of BALB/c mice, indicating its genotoxic risk at high doses [331]. Evaluating whether TQ-nanostructured lipid carrier (TQ-NLC) shows any toxic effect, Yazan and the team observed that after TQ-NLC administration (25 mg/kg, intravenous route), there was no visible change in the body weight, food intake, organ-to-body weight ratio, and hematological, biochemical, and histopathological profile, although an inflammation was marked at the injection site in rat tails [325]. Thus, nanocarrier-based TQ delivery may minimize toxicity although it was not clear why more bioavailable TQ caused relatively less toxicity.

Evaluating the effect of TQ on pregnant rats and embryo-fetal development, pregnant rats were given a single injection (15, 35, and 50 mg/kg, i.p.) of TQ on the 11th and 14th day of gestation. It was observed that TQ at 15 mg/kg showed no adverse effect on maternal health and embryo–fetal development. However, TQ at 35 mg/kg caused both maternal and embryonic toxicities on 11th day. Moreover, following administration of TQ at 50 mg/kg, there was a significant decrease in maternal body weight and complete fetal resorption on 11th day of gestation. These findings suggest that embryonic development may be impaired, particularly when pregnant mothers are exposed to TQ at doses of 35 and 50 mg/kg in the second trimester of pregnancy [332]. However, phytovagex (a black cumin formulation indicated for vaginal fungal infection) and black cumin oil applied to pregnant rats as an intravaginal pessary in the first half of pregnancy did not cause any adverse effect on the duration of pregnancy and health parameters of the offspring [337,338]. 

Several toxicological studies reported lethal dose (LD50) of TQ. Like a sublethal toxic dose, the lethal dose of TQ also varies with the route of administration where LD50 of intraperitoneal TQ was much lower than that of the oral route (Figure 7c). For example, in mice, LD50 of TQ was found to be 104.7 mg/kg (89.7–119.7) and 870.9 mg/kg (647.1–1094.8), respectively, following intraperitoneal and oral administration. The LD50 of TQ also varies with animal models used. In contrast to mice, LD50 in rats was 57.5 mg/kg (45.6–69.4) and 794.3 mg/kg (469.8–1118.8), respectively, when administered intraperitoneally and orally [333]. Moreover, the LD50 values of TQ after intraperitoneal injection and oral gavages were 10–15 times and 100–150 times greater than its effective doses for anti-inflammatory, antioxidant, and anti-cancer activities [333]. These observations lead us to conclude that the therapeutic window of TQ is relatively wide and that the oral TQ is safer than that given intraperitoneally. Oral TQ may undergo biotransformation to produce less toxic metabolites in the gastrointestinal tract or become metabolized in the liver and produce dihydrotymoquinone. Whereas, intraperitoneal administration of TQ facilitated its transport into the systemic circulation and amplified its toxicity [330,334]. The minimum or zero toxicity and wider therapeutic window of black cumin and TQ as evident by the aforementioned literature endorse its long-term traditional use as food and medicine. Although the safety of the use of black cumin by humans has been supported by conventional knowledge, future studies with human subjects are warranted to unequivocally confirm the safety issues related to the therapeutic use of black cumin and TQ.

## 10. Concluding Remarks and Future Perspectives

Existing evidence substantiates pharmacological benefits of black cumin and TQ, covering almost every physiological system. However, the therapeutic potential of black cumin against various diseases has not been investigated with equal emphasis as some chronic diseases, such as cancer, neurological disorders, and metabolic syndromes, have priority over other diseases. Antioxidant, anti-inflammatory, antiapoptotic, and immunomodulatory properties are the major pharmacological attributes of black cumin and TQ that contribute to their potential health benefits against a wide range of disease conditions. As evident in several studies, black cumin and TQ can also function as efficient natural antidotes, protecting against toxicities in various organs, including the brain, kidney, lung, liver, heart, gastrointestinal tracts, and reproductive system. Because of their potent chelating capacity, black cumin and its constituents can eliminate a variety of toxins that are commonly contaminated or adulterated in food. Furthermore, black cumin and TQ could potentially mitigate side effects of several existing drugs that are used for cancer and other human diseases.

In many cases, TQ has often been credited for the pharmacological effects of black cumin, regardless of the research focused on its health benefits. However, some studies have presented important findings on the significant health benefits caused by other phytochemicals, such as thymohydroquinone [339], thymol [340], and carvacrol [341], suggesting that compounds beyond TQ also deserve attention.

Among the signaling pathways that are targeted by black cumin and TQ, the most significant pathways are those that are associated with the antioxidant defense system, anti-inflammation and immune system, anti-apoptosis and cell survival system, autophagy, and energy metabolism. Although the underlying mechanisms of many systemic effects are well demonstrated, some effects are still in their infancy and, thus, require further elucidation. As black cumin and its constituents target many cell signaling systems producing pharmacological effects in almost every organ/system, it may comply with the multidrug–multitarget concept. Integrated system pharmacology and computational approach can, therefore, be employed to gain an in-depth insight into the pharmacological effects of black cumin and TQ, particularly against those diseases that are interlinked, such as chronic metabolic diseases, cancer, and neurodegenerative disorders.

Apart from individual effects, when applied in combination with other compounds or natural products, black cumin or TQ potentiates their biological activity, as well as synergizes the effects, reducing the dose size of the concurrently used drugs and minimizing the likely toxicity. However, the pharmacological interactions that occur when used concomitantly with other compounds have been highlighted so that further research in the future does not overlook this critical issue in the path to the development of a successful drug.

Although significant progress has been made in terms of pharmacological benefits of black cumin and TQ, this ancient medicine is still far from its clinical application. Since the therapeutic use of TQ is limited by its poor bioavailability, strategies, such as purposeful modification of its structure without hindering biological activity, may improve its bioavailability. Several recent studies employed the nanoparticle synthesis of black cumin and TQ to improve their bioavailability and pharmacological effects. However, most of the investigation was performed at the preclinical level, thus, it is essential to carry out the human trial to translate the findings into clinical use. Although black cumin and TQ show limited/no toxicity, this information was mostly based on preclinical studies. Thus, an extensive human trial is warranted to validate pharmacological and toxicological profiles for future clinical use.

We anticipate that information from this critical review would be capitalized for the future development of black cumin-based potential therapeutic agents to be applied for various diseases and the formulation of popular functional foods in order to promote optimum health and reduce the risk of chronic disease.

## Figures and Tables

**Figure 1 nutrients-13-01784-f001:**
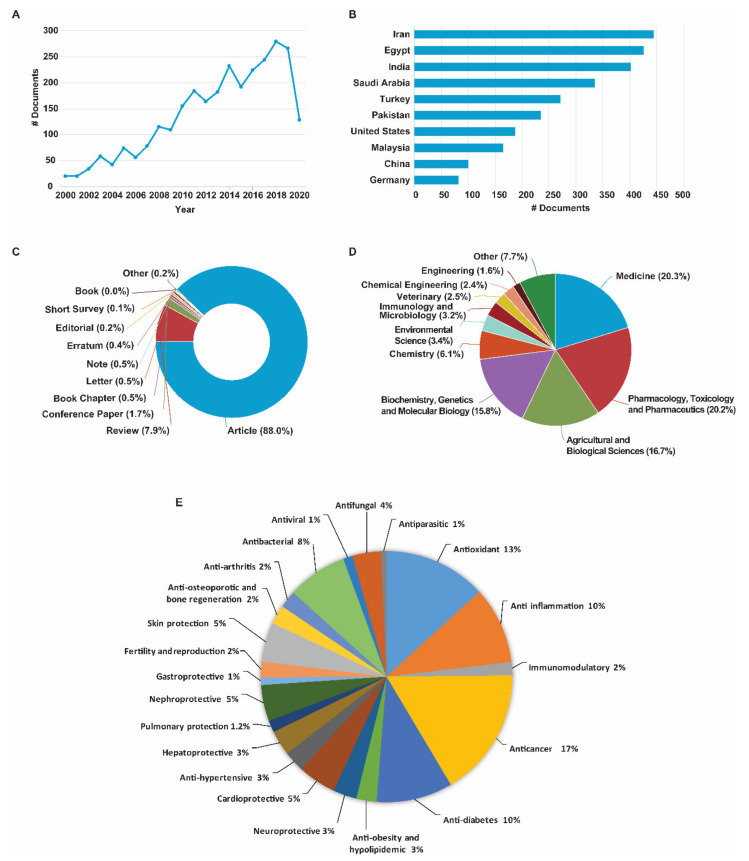
Research trends in black cumin. (**A**) Yearly appearance of publications. (**B**) Top 10 countries with the highest number of publications. (**C**) Document-wise proportional rate of publications. (**D**) Proportional rate of publications according to research areas. (**E**) Proportional rate of publications according to pharmacological effects. The data were retrieved from the Scopus database in June 2020.

**Figure 2 nutrients-13-01784-f002:**
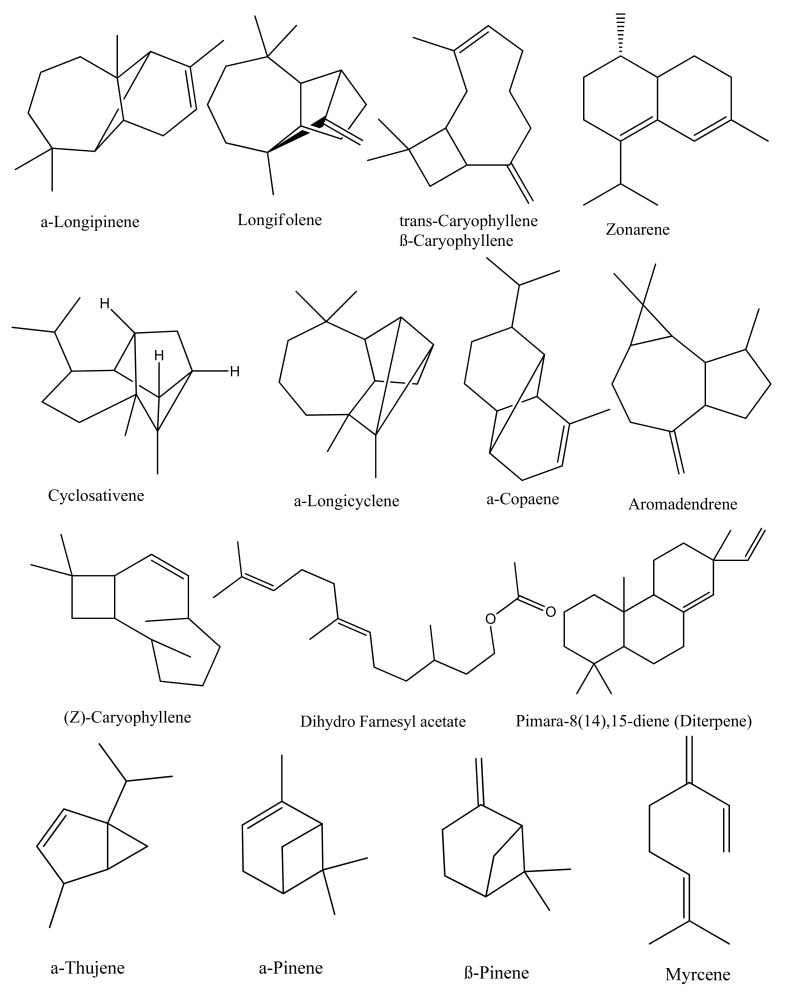
Chemical structure of (**A**) terpenes and terpenoids, (**B**) phytosterols, (**C**) alkaloids, (**D**) tocols, and (**E**) polyphenols.

**Figure 4 nutrients-13-01784-f004:**
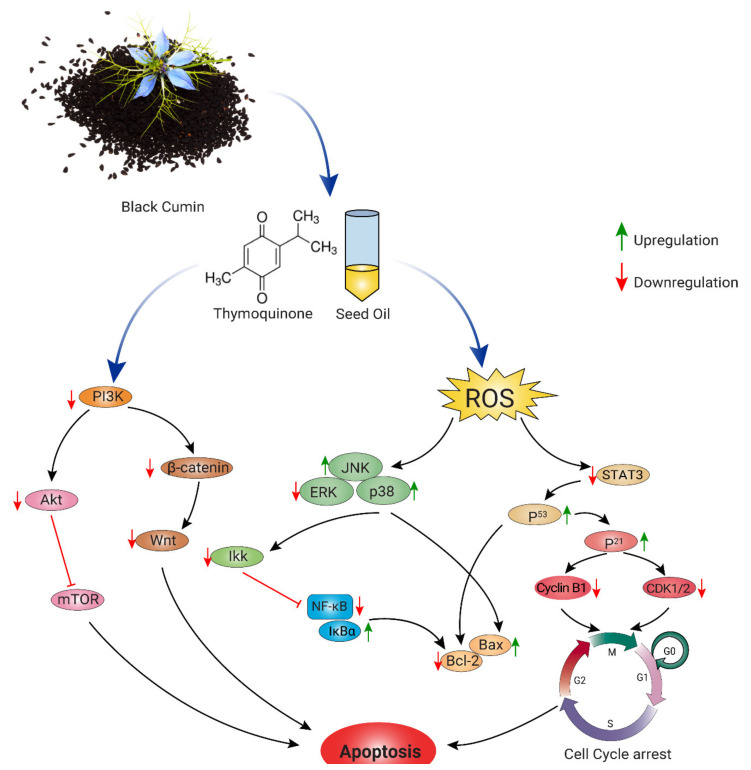
Plausible anticancer mechanism of black cumin, TQ, and essential oil: triggering cellular apoptosis and cell cycle arrest by targeting multiple signaling pathways. Black cumin and its components provoked cancer-cell specific apoptosis via altering several signaling cascades, including PI3K/Akt/mTOR, Wnt/β-catenin, and NF-κB signaling. Black cumin and its components also caused DNA damage that involves several mechanisms such as ROS induction and subsequent increase in oxidative stress and mitochondrial dysfunction which eventually increase Bax/Bcl-2 ratio through c-Jun N-terminal kinase (p-JNK) pathway. Black cumin also suppressed Cyclin B1 and CDK1/2 expression through regulating STAT3 and MAPK pathways and caused cell cycle arrest. ROS, reactive oxygen species; MAPK, mitogen-activated protein kinase; STAT3, signal transducer and activator of transcription-3; JNK, c-Jun N-terminal kinase; ERK, extracellular signal-regulated kinase; NF-κB, nuclear factor kappa-light-chain-enhancer of activated B cells; CDK1/2, cyclin-dependent kinase 1/2; Cyclin B1, regulatory protein of maturation-promoting factor.

**Figure 5 nutrients-13-01784-f005:**
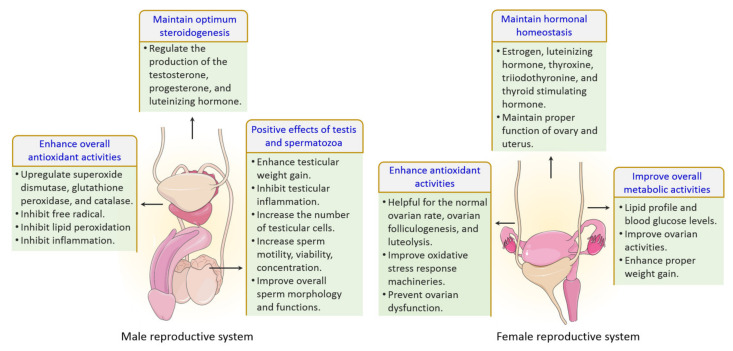
Hypothetical illustration showing black cumin action mechanism on reproduction. At the molecular level, black cumin exhibits its beneficial effects on reproduction via three major pathways: (1) adjust hormonal homeostasis, (2) enhance antioxidant capacity of reproductive tissue/cells, and (3) facilitate proper growth and maturation of germ cells and associated organs. A more detailed description of black cumin action can be found in the main text.

**Figure 6 nutrients-13-01784-f006:**
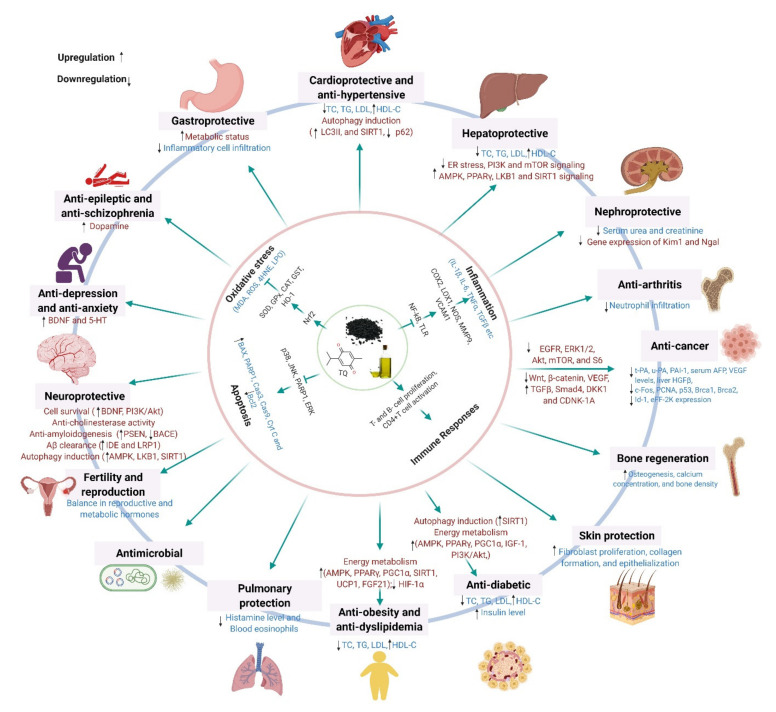
Comprehensive molecular mechanism of black cumin and TQ-mediated pharmacological actions. The general pharmacological effects are manifested by their capacity to attenuate oxidative stress by activating the antioxidant defense system (Nrf2 signaling), inhibit inflammation by activating anti-inflammatory signaling (NF-κB and TLR signaling), induce immunity by modulating innate and adaptive immune components, prevent apoptosis by upregulating pro-survival signals and downregulating pro-apoptotic signals (PI3K/Akt, JNK, and mTOR signaling). Other significant molecular mechanisms include induction of autophagy (SIRT1 signaling), priming of energy metabolism (AMPK-SIRT1-PGC-1α and PPARγ signaling), activation of growth factor signaling (PI3K/Akt signaling), and enhancement of protein clearance by upregulating LRP1. Through employing multiple of these pharmacological mechanisms, black cumin and TQ exerted their health benefits including protection against metabolic (obesity, dyslipidemia, and diabetes), cardiovascular, digestive, renal, hepatic, osteogenic, respiratory, reproductive, neurological and mental disorders, and various types of cancer.

**Table 3 nutrients-13-01784-t003:** Comprehensive summary on the anti-obesity and anti-dyslipidemia effects of black cumin.

Treatment with Doses	Experimental Model	Major Findings (Including Molecular Changes)	References
Seed powder (500 mg/kg)	Hyperlipidemic albino rats	↓Serum cholesterol, Triglycerides, LDL levels; ↑HDL levels	[122]
Seed (1 g) (+ 1 tablespoon of honey + 10 mg atorvastatin)	Hyperlipidemic patients with a history of smoking habit	↓TC, LDL, TG and ↑HDL	[123]
Seed extract (treated with vinegar for a week to make 500 mg capsule)	Double-blind, randomized placebo-controlled trial among menopausal women with metabolic syndrome	↓LDL, TG, TC, and fasting blood sugar	[124]
NSO and hypoglycemic drug (1.5 mL and 3 mL)	Single-blind, randomized controlled trial on outpatients with metabolic syndrome	↓HbA1c levels	[125]
Seed powder (2 g)	Patients with Hashimoto’s thyroiditis	↑Serum TAC, SOD, HDL and ↓MDA, VCAM-1, BMI, LDL, and TG; no significant change in GPX	[126]

**Table 4 nutrients-13-01784-t004:** Comprehensive summary on the antidiabetic effects of black cumin.

Treatment with Doses	Experimental Model	Major Findings (Including Molecular Changes)	References
Silver nanoparticles prepared from NSO	In vitro biochemical assay	↓α-amylase activity	[139]
Seed extract (125 and 250 mg/kg)	Alloxan-induced diabetic rats	↓Glucose and MDA levels; ↑SOD and GPx; ↑diameter and amount of Langerhans islet cells	[128]
Seed extract (2 g/kg)	Alloxan-induced diabetic mice	↓Blood glucose, TG, T-cholesterol, LDL-c, and TBARS; ↑HDL-c	[129]
Seed extract (20% or 40%)	STZ-induced diabetic rats	Shortens duration of wound healing (15 days in 40% seed extract-treated vs 27 days in untreated)	[131]
Seed extract (100, 200, and 400 mg/kg)	STZ-induced diabetic rat	↓Serum glucose and lipids; ↑AIP; ↑eNOS expression; ↓VCAM-1 and LOX-1 expressions	[130]
Green synthesis of silver nanoparticles using seed extract (200 mg/kg)	STZ-induced model of diabetic neuropathy in rats	↓Glucose and AGE and aldose reductase level; ↓TNF-α, NF-κB and S100B; ↓MDA and NO; ↑SOD and GSH; ↑TKr A; ↑nitrotyrosine	[132]
NSO (400 mg/kg)	STZ -induced diabetic rats	↓Myositis, hyaline degeneration, and Zenker’s necrosis; ↑Bcl-2 expression	[134]
NSO (0.2 mg/kg)	STZ-induced diabetic rats	↓Blood glucose; ↓Bax and Caspase 3 expression in aortic medial layer	[133]
NSO (100 mg/ Kg)	STZ-induced diabetic rats	↑Insulin-like growth factor-1 and phosphoinositide-3 kinase; ↓ADAM-17; ↓blood glucose level, lipid profile, TBARS, NO, serum insulin/insulin receptor ratio, and the tumor necrosis factor	[135]
NSO (2 mL/kg)	STZ-induced diabetic rats	↓FBG; ↑insulin levels; ↑pancreatic and hepatic CAT and GSH; ↑insulin immunoreactive parts % and mean pancreatic islet diameter	[136]
NSO (2 mL/kg) and TQ (50 mg/kg)	STZ-induced model of diabetic nephropathy	↓Albuminuria and the kidney weight/body weight ratio; ↑podocyte-specific marker; ↓collagen IV, TGF-β1 and VEGF-A	[137]
NSO (2.5 mL/kg)	Alloxan-induced diabetic rabbits	↓CAT activity, TC, TGs, LDL- cholesterol and VLDL- cholesterol levels, serum blood glucose levels and lipid contents; ↑mean body weight, HDL- cholesterol, vitamin C levels and normalized bilirubin levels	[138]
NSO (2.5 mL/kg)	Randomized clinical trial on patients with diabetic nephropathy accompanying chronic kidney disease	↓Blood glucose, serum creatinine, blood urea, and 24 h total urinary protein levels; ↑glomerular filtration rate, 24 h total urinary volume, and hemoglobin level	[140]
NSO (1 g as two capsules)	Double-blind randomized clinical trial on T2DM patients	↓Lipid profile and glucose level, C-reactive protein level, and lipid peroxidation	[141]
Seed capsules (2 g daily)	Single-blind, nonrandomized controlled clinical trials on T2DM patients	↓TC, LDL-C, TC/HDL-C and LDL-C/HDL-C ratios; ↑serum HDL-C; ↓DBP, MAP and HR	[142]

**Table 7 nutrients-13-01784-t007:** Comprehensive summary on the pulmonary protective effects of black cumin.

Treatment with Doses	Experimental Model	Major Findings (Including Molecular Changes)	References
Seed extract (100, 200, 400 mg/kg, i.p. for 16 days started 2 days prior to LPS injection)	LPS-induced lung damage in rats	↓Inflammatory leukocytes and oxidative stress markers in the blood and bronchoalveolar fluid; ↓inflammatory cytokines (IFN-γ, TGF-β1, PGE2); ↑IL-4 in bronchoalveolar fluid	[185]
Seed extract (500 mg/kg per day)	Bleomycin-induced lung fibrosis (BMILF) in rats	↓Pulmonary inflammation and fibrosis; ↓hydroxyproline concentration in pulmonary tissue ↓lipid peroxidation; ↑CAT	[186]
NSO (1 mL/kg per day, p.o. for 50 days)	BMILF in rats	↓Inflammatory index and fibrosis score; ↑urinary secretion of histidine fumarate, allantoin and malate	[188]
TQ (10 and 20 mg/kg, p.o. for 28 days)	BMILF in rats	Recovers weight loss; ↓MMP-7 expression; ↓apoptotic markers (caspase 3, Bax, and Bcl-2); ↓fibrotic changes (TGF-β and hydroxyproline in lung tissues); ↑Nrf2, HO-1 and ↓TGF-β expression	[187]
TQ (i.p. daily for the last 21 days of a total of 90 days exposure to cigarette smoke)	Cigarette smoke-exposed chronic obstructive pulmonary disease (COPD) rats	↓IL-8 levels; ↓↑apoptosis (hence, it requires appropriate dosing of TQ while long term TQ or DMSO exposure might have toxicity)	[189]
NS seed extract (+ onion extracts) for 18 weeks	Nicotine-induced lung damage in SD rats	↓MDA level; ↑SOD, CAT, and GSH to a normal level	[190]
NSO (1 mL/kg of NSO through gastric gavage for 28 days one hour before CdCl_2_ administration)	Cadmium-induced lung damage in adult male rats	↓Cd-induced lung damage with minimal histopathological changes in lung architecture	[191]
Seed extract (0.1–0.5 mg/mL)	Wistar rat peritoneal mast cells	↓Histamine level, no toxicity on mast cells	[192]
TQ-rich oily preparations	Human T lymphocyte, monocyte and A549 human lung epithelial cells (in vitro models of asthma-related mediators of inflammation)	↓IL-2, IL-6, and PGE_2_ in T-lymphocytes; ↓IL-6 and PGE_2_ in monocytes; ↑ PGE_2_ in A549 cells	[41]
Black cumin supplement (NS-1, NS-2 =1, 2 g/day respectively, for 3 months)	Randomized single-blind, placebo-controlled clinical trial with asthma patients	↑Peak expiratory flow by 25–75%; ↓fractional exhaled nitric oxide and serum IgE; ↑serum IFN-γ,Improvement in asthma control test score	[193]
NSO capsules (500 mg, twice daily for 4 weeks)	Randomized single-blind, placebo-controlled clinical trial with asthma patients	↓Blood eosinophils; ↑asthma control with an overall improvement in pulmonary function	[194]

**Table 8 nutrients-13-01784-t008:** Comprehensive summary on the gastroprotective effects of black cumin.

Treatment with Doses	Experimental Model	Major Findings (Including Molecular Changes)	References
NSO (2 mL/kg BW, p.o.)	Cisplatin (CP)-induced gastrointestinal dysfunction in rats	↓BBM enzyme activities and BBMV; ↓carbohydrate metabolism enzyme activities and enzymatic as well as non-enzymatic antioxidant parameters in the intestine; ↑intestinal redox and metabolic status; restored BBM integrity	[196]
TQ (1.5 mg/kg BW)	CP-induced gastrointestinal damage in rats	↓CP induced specific activities of BBM marker enzymes; restoring the redox and metabolic status of intestinal mucosal tissue and preserving intestinal histoarchitecture	[197]
NSO (2 mL/kg BW, p.o.) and TQ (1.5 mg/kg BW, p.o.), for 14 days	CP-induced intestinal toxicity in rats	↑BBM enzyme activities; ↑carbohydrate metabolism enzymes and the enzymatic as well as non-enzymatic parameters of antioxidant defense system in the intestinal mucosa	[198]
Pectic polysaccharide of seed (200 mg/kg BW for 10 days)	Acetic acid-induced gastric ulcers in rats.	↑Gastric mucin content, Cox-2, PGE2, ERK-2, MMP-2; ↓MMP-9 levels; ↓H^+^/K^+^-ATPase and free radical-mediated oxidation and cellular damages that improved speed and quality of ulcer healing	[199]
TQ (5 mg/kg)	Fasting-induced rabbit gastric ulcer model	↓ Volume and the total acidity of gastric secretion	[201]
NSO (0.2 mL/kg/day for 5 days)	Male Wistar albino rats with ischemia/ reperfusion injury (colonic anastomosis model)	↑Serum hydroxyproline levels, IL-6 level; ↓tissue levels and serum levels of TNF-α; ↓ edema and inflammatory cell infiltration; significant difference in Chiu classification	[202]
Dosin (6 g/day of black cumin seed and 12 g/day of honey)	Patients with *H. pylori* infection	Negative UBT after intervention; ↓median and interquartile range of total dyspepsia symptoms	[203]
Seed powder (2 g for 6 weeks)	Patients with moderate ulcerative colitis	↓ Stool frequency	[204]

**Table 10 nutrients-13-01784-t010:** Comprehensive summary on the bone regenerative effects of black cumin.

Treatment with Doses	Experimental Model	Major Findings (Including Molecular Changes)	References
Seed extract	Socket healing in rabbits	↑Active bone formation, thick trabeculae with highly vascular bone marrow and large numerous osteocytes	[235]
Seed extract	Calvarial defected ovariectomy-induced osteoporosis in rat models	↑Bone formation; ↑bone healing	[234]
TQ (10 mg/kg/day, p.o.)	Tibias defected male rats	↑Ratio of new bone per total defect area and new bone trabeculae lined by active osteoblasts; ↑capillary intensity in the defect area;↑osteogenesis	[236]
NSO	Dental pulp MSCs isolated from 15–20 years old human patients from third molars	↑Calcium concentrations	[237]
Seeds	Clinical trial with healthy patients (for evaluation of Topical NS application on Delayed Dental Implant	↑Bone density after six months	[238]

**Table 11 nutrients-13-01784-t011:** Antidotal effects of black cumin and its constituents.

Treatment with Doses	Model of Toxicity	Mechanism of Antidote Action	References
TQ (2.5, 5, 10, 15, and 20 μM)	2-tert-Butyl-4-hydroquinone-induced cytotoxicity in human umbilical vein endothelial cells	Anti-apoptotic and anti-DNA and chromatin fragmentation effects	[288]
TQ (500 μM)	Chromium-induced in vitro toxicity	Act as a chelating agent	[283]
Ethanolic seed extract (0–25 µL)	FeSO_4_-induced toxicity in rat	Act as a chelating agent; Antioxidant function	[282]
TQ (10 mg/kg daily for one month)	Malathion-induced toxicity in rats	Antioxidant function	[284]
TQ (10 mg/kg BW, p.o., once daily for 28 days)	Fipronil (phenylpyrazole insecticide)-induced oxidative injury in rats	Antioxidant function	[285]
TQ (2.5, 5, 10 mg/kg/day)	Diazinon-induced toxicity in rats	Antioxidant function	[286]
TQ (20 mg/kg for three weeks)	Samsun ant (*Pachycondyla sennaarensis*) venom-induced acute toxic shock in male rats	Antioxidant and antiallergic functions	[287]

## Data Availability

Not applicable.

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
