# Peer review of "Black Cumin (*Nigella sativa* L.): A Comprehensive Review on Phytochemistry, Health Benefits, Molecular Pharmacology, and Safety"

_nutrients, 2021, doi:10.3390/nu13061784_

Round 1
Reviewer 1 Report
Functional food potentials of black cumin (Nigella sativa L.): health benefits, pharmacological insights, and safety- Authors, Abdul Hannan et al.
I would like to congratulate the authors of this review, as it has been a special delight for me to read this work, since I consider that its authors have done an excellent job of collecting and organizing data. For this reason, I believe that this review can be accepted for publication, however, I request that the authors make a special effort to carry out a minor review since in my opinion they should correct three formal aspects and one in relation to the substance of the content.
Formal aspects.
1º.- The title of the work is far below the content of the review, it could probably be achieved using practically the same words, but changing them.
2º.- The authors use the phrase “Per os” as PO to indicate the oral intake of the compound, oil or seed, however, they also use the terms oral or oral administration. I think that the authors should always use the same term or phrase, but if they finally decide to leave all these terms, they should clarify the acronym PO in the first moment that it appears in the tables and not later as it happens in the current writing of the work
3rd and very important. - A table that is presented in up to 9 pages in a row is very complicated for readers to follow its usefulness as the text is read. The authors should make a special effort to divide them using part of their information in the text of the work separating the nutritional, molecular, physiological and pharmacological effects.
Background aspect
Taking into account that one of the pharmacological aspects of black cumin is its role against epilepsy, the authors have done minimal work in the study of this behavior and should go a little deeper.
Author Response
RESPONSE TO REVIEWER COMMENTS
Manuscript ID: nutrients-1202406
Title: Black cumin (Nigella sativa L.): a comprehensive review on phytochemistry, health benefits, molecular pharmacology, and safety
We greatly appreciate respected Reviewer for the valued comments which have surely enriched our manuscript. In the following text, we respond to the reviewer’s comments point by point and highlight all changes in the manuscript file.
Point by point response to (#1) Reviewer’s comments
General comments
I would like to congratulate the authors of this review, as it has been a special delight for me to read this work, since I consider that its authors have done an excellent job of collecting and organizing data. For this reason, I believe that this review can be accepted for publication, however, I request that the authors make a special effort to carry out a minor review since in my opinion they should correct three formal aspects and one in relation to the substance of the content.
Author response: We appreciate the reviewer for this encouraging comment.
Formal aspects
1º.- The title of the work is far below the content of the review, it could probably be achieved using practically the same words, but changing them.
Author response: We thank reviewer for this suggestion and accordingly have modified the manuscript title.
2º.- The authors use the phrase “Per os” as PO to indicate the oral intake of the compound, oil or seed, however, they also use the terms oral or oral administration. I think that the authors should always use the same term or phrase, but if they finally decide to leave all these terms, they should clarify the acronym PO in the first moment that it appears in the tables and not later as it happens in the current writing of the work
Author response: We thank reviewer for this concern. We used the phrase “Per os” abbreviated as PO to indicate both oral intake and oral administration (in common, oral route). We elaborated the acronym PO in its first appearance (Page 14) and also in the table footnote (Table 1A).
3rd and very important. - A table that is presented in up to 9 pages in a row is very complicated for readers to follow its usefulness as the text is read. The authors should make a special effort to divide them using part of their information in the text of the work separating the nutritional, molecular, physiological and pharmacological effects.
Author response: We thank the reviewer for this concern. In fact, Table 1 was originally not designed according to nutritional, molecular, physiological, and pharmacological effects. Now, we have split Table 1 into Table 1A-J based on the pharmacological effects of black cumin against each disease condition so that readers can follow the table and the relevant text. Hope these changes have sufficiently addressed the reviewer's concern.
Background aspect
Taking into account that one of the pharmacological aspects of black cumin is its role against epilepsy, the authors have done minimal work in the study of this behavior and should go a little deeper.
Author response: We appreciate reviewer for this recommendation. Accordingly, we included a separate subsection highlighting epilepsy on page 20. To minimize manuscript size, we only referred to other neurological problems as these were reviewed elsewhere [1].
6.4.5. Protection against anxiety and depression, epilepsy, schizophrenia and other miscellaneous neurological problems
As shown in Table 1A, black cumin and TQ have also shown promises as an antidepressant and anxiolytic [2-5] and anti-schizophrenic [6] agents. In several animal models of epilepsy, black cumin and TQ have been shown to reduce convulsion and improve memory performance. Vafaee and colleagues reported that oral administration of hydroalcoholic seed extract attenuated convulsion and improved memory performance in pentylenetetrazole (PTZ)-induced seizure model through modulating redox status [7]. In another study, black cumin and probiotic supplementation conferred protection against seizure, seizure-induced cognitive impairment, and hippocampal long-term potentiation in PTZ-induced kindled rats [8]. NSO exhibited anticonvulsant effects in electroshock-induced seizures in rats [9]. In their two successive studies, Shao and the team reported that TQ attenuated convulsion and improved learning and memory function in Lithium-pilocarpine induced models of status epilepticus via anti-inflammatory (NF-κB) [10] and antioxidant (Nrf2/HO-1) [11] signaling pathways. TQ was also shown to mitigate various neuropathic pain [12, 13] because of the antioxidant, anti-inflammatory, anti-apoptosis and neurotrophic properties of black cumin.
Reference
- Jakaria, M.; Cho, D. Y.; Haque, M. E.; Karthivashan, G.; Kim, I. S.; Ganesan, P.; Choi, D. K., Neuropharmacological potential and delivery prospects of thymoquinone for neurological disorders. Oxidative Medicine and Cellular Longevity 2018, 2018.
- Ahirwar, D.; Ahirwar, B., Antidepressant effect of nigella sativa in stress-induced depression. Research Journal of Pharmacy and Technology 2020, 13, (4), 1611-1614.
- Alam, M.; Zameer, S.; Najmi, A. K.; Ahmad, F. J.; Imam, S. S.; Akhtar, M., Thymoquinone Loaded Solid Lipid Nanoparticles Demonstrated Antidepressant-Like Activity in Rats via Indoleamine 2, 3- Dioxygenase Pathway. Drug Research 2020, 70, (5), 206-213.
- Farh, M.; Kadil, Y.; Tahri, E. H.; Abounasr, M.; Riad, F.; El Khasmi, M.; Tazi, A., Evaluation of anxiolytic, antidepressant, and memory effects of Nigella sativa seeds oil in rat. Phytotherapie 2017, 1-9.
- Beheshti, F.; Norouzi, F.; Abareshi, A.; Anaeigoudari, A.; Hosseini, M., Acute administration of Nigella sativa showed anxiolytic and anti-depression effects in rats. Current Nutrition and Food Science 2018, 14, (5), 422-431.
- Khan, R. A.; Najmi, A. K.; Khuroo, A. H.; Goswami, D.; Akhtar, M., Ameliorating effects of thymoquinone in rodent models of schizophrenia. African Journal of Pharmacy and Pharmacology 2014, 8, (15), 413-421.
- Vafaee, F.; Hosseini, M.; Hassanzadeh, Z.; Edalatmanesh, M. A.; Sadeghnia, H. R.; Seghatoleslam, M.; Mousavi, S. M.; Amani, A.; Shafei, M. N., The effects of Nigella sativa hydro-alcoholic extract on memory and brain tissues oxidative damage after repeated seizures in rats. Iranian Journal of Pharmaceutical Research 2015, 14, (2), 547-557.
- Tahmasebi, S.; Oryan, S.; Mohajerani, H. R.; Akbari, N.; Palizvan, M. R., Probiotics and Nigella sativa extract supplementation improved behavioral and electrophysiological effects of PTZ-induced chemical kindling in rats. Epilepsy and Behavior 2020, 104.
- Bepari, A.; Parashivamurthy, B. M.; Niazi, S. K., Evaluation of the effect of volatile oil extract of Nigella sativa seeds on maximal electroshock-induced seizures in albino rats. National Journal of Physiology, Pharmacy and Pharmacology 2017, 7, (3), 273-284.
- Shao, Y.; Feng, Y.; Xie, Y.; Luo, Q.; Chen, L.; Li, B.; Chen, Y., Protective Effects of Thymoquinone Against Convulsant Activity Induced by Lithium-Pilocarpine in a model of Status Epilepticus. Neurochemical research 2016, 41, (12), 3399-3406.
- Shao, Y. Y.; Li, B.; Huang, Y. M.; Luo, Q.; Xie, Y. M.; Chen, Y. H., Thymoquinone attenuates brain injury via an antioxidative pathway in a status epilepticus rat model. Translational Neuroscience 2017, 8, (1), 9-14.
- Çelik, F.; Göçmez, C.; Karaman, H.; Kamaşak, K.; Kaplan, İ.; Akıl, E.; Tufek, A.; Guzel, A.; Uzar, E., Therapeutic Effects of Thymoquinone in a Model of Neuropathic Pain. Current Therapeutic Research 2014, 76, 11-16.
- Amin, B.; Taheri, M. M.; Hosseinzadeh, H., Effects of intraperitoneal thymoquinone on chronic neuropathic pain in rats. Planta Med 2014, 80, (15), 1269-77.
Reviewer 2 Report
The manuscript presents functional properties of black cumin. The review is well organised and precisely described current state of the art in health p[ropereties, pharmacological insights and safety of black cumin and its components. The review is woth of publishing in the Nutrients and I reccomend to accept it in submitted form.
Author Response
RESPONSE TO REVIEWER COMMENTS
Manuscript ID: nutrients-1202406
Title: Black cumin (Nigella sativa L.): a comprehensive review on phytochemistry, health benefits, molecular pharmacology, and safety
General comments
The manuscript presents functional properties of black cumin. The review is well organized and precisely described current state of the art in health properties, pharmacological insights and safety of black cumin and its components. The review is worth of publishing in the Nutrients and I recommend to accept it in submitted form.
Author response: We greatly appreciate our valued reviewer for this comment that encourages us to contribute more to science.